# Magnetically driven capsules with multimodal response and multifunctionality for biomedical applications

Yuxuan Sun [1,2,7], Wang Zhang[1,2,7], Junnan Gu[3], Liangyu Xia[1,2], Yinghao Cao[4], Xinhui Zhu[1,2], Hao Wen[1,2], Shaowei Ouyang[1,2], Ruiqi Liu[1,2], Jialong Li[1,2], Zhenxing Jiang[3], Denglong Cheng[3], Yiliang Lv[1,2], Xiaotao Han[1,2], Wu Qiu[5], Kailin Cai[3], Enmin Song[6], Quanliang Cao [1,2] ✉ & Liang Li[1,2] ✉

Untethered capsules hold clinical potential for the diagnosis and treatment of gastrointestinal diseases. Although considerable progress has been achieved recently in this field, the constraints imposed by the narrow spatial structure of the capsule and complex gastrointestinal tract environment cause many open-ended problems, such as poor active motion and limited medical functions. In this work, we describe the development of small-scale magnetically driven capsules with a distinct magnetic soft valve made of dual-layer ferromagnetic soft composite films. A core technological advancement achieved is the flexible opening and closing of the magnetic soft valve by using the competitive interactions between magnetic gradient force and magnetic torque, laying the foundation for the functional integration of both drug release and sampling. Meanwhile, we propose a magnetic actuation strategy based on multifrequency response control and demonstrate that it can achieve effective decoupled regulation of the capsule's global motion and local responses. Finally, through a comprehensive approach encompassing ideal models, animal ex vivo models, and in vivo assessment, we demonstrate the versatility of the developed magnetic capsules and their multiple potential applications in the biomedical field, such as targeted drug delivery and sampling, selective dual-drug release, and light/thermal-assisted therapy.

The gastrointestinal (GI) tract is one of the largest hormone-producing organs in the body[1], which induces the most common clinical disorders such as inflammation, ulcers, hemorrhage, infections and cancers[2]. According to the cancer statistics presented in 2023[3], colorectal and stomach cancers that are malignant tumors of the GI tract rank among the top three in terms of cancer incidence, posing a severe threat to human health. As the recovery and survival rates of GI diseases decrease significantly with the prolongation of the onset of the disease, the development of efficient early medical diagnosis-screening techniques and equipment has excellent significance for the protection of GI health[4,5]. However, both traditional flexible endoscopy and capsule endoscopy have considerable limitations and

[1]Wuhan National High Magnetic Field Center, Huazhong University of Science and Technology, Wuhan 430074, China. [2]State Key Laboratory of Advanced Electromagnetic Technology, Huazhong University of Science and Technology, Wuhan 430074, China. [3]Department of Gastrointestinal Surgery, Union Hospital, Tongji Medical College, Huazhong University of Science and Technology, Wuhan 430022, China. [4]Cancer Center, Union Hospital, Tongji Medical College, Huazhong University of Science and Technology, Wuhan 430022, China. [5]School of Life Science and Technology, Huazhong University of Science and Technology, Wuhan 430074, China. [6]School of Computer and Technology, Huazhong University of Science and Technology, Wuhan 430074, China. [7]These authors contributed equally: Yuxuan Sun, Wang Zhang. ✉e-mail: quanliangcao@hust.edu.cn; liangli44@hust.edu.cn

need urgent innovative breakthroughs: conventional flexible endoscopy can be used for integration of diagnosis and treatment, but patients experience intense discomfort as well as contraindications[6–8]. Although capsule endoscopy can enable wire-free, non-invasive, and painless diagnostic procedures, its movement often relies on peristalsis in the intestines and is mainly restricted to imaging purposes. This makes it unsuitable for medical procedures like sampling or targeted drug delivery[9,10]. The field of micro-robotics is currently evolving rapidly, with its primary goal being to fulfil a wide range of biomedical applications within the human body[11–14]. Combining traditional capsules with modern micro-robotics can lead to promising diagnostic and therapeutic approaches that may provide innovative solutions to the aforementioned problems and challenges. Published reports present an investigation of diverse actuation methods, mainly classified as passive and active. The passive capsules mainly use pH[15] and temperature as actuation ways. These systems are usually simple in structure but have difficulty controlling variables such as the release rate, target location and number of dosages[16]. Compared with the passive capsules, the active ones can provide richer biomedical functions, such as targeted therapy and multi-source therapy[17].

Representative actuation methods of existing capsules include pneumatic or hydraulic[18], electromechanical[19–23], shape memory alloys[24–27], and magnetic actuation[28–35]. Out of these methods, the magnetic actuation is more straightforward[36] and holds potential for size downscaling[37], as it eliminates the need for internal on-board control and power supply devices. Furthermore, it has significant advantages such as non-contact, high controllability, and good penetration performance[38,39]. Therefore, it is believed to be one of the safest and most ideal methods for capsules. An encouraging representative work carried out in the field of GI disease diagnosis has been the development of magnetic capsule endoscopy[40,41], which has injected strong confidence into the development and biomedical applications of magnetically driven capsules.

However, the application scenarios of the existing magnetic capsule endoscopy are significantly limited, because it is mainly used for image acquisition and transmission without additional diagnosis and treatment functions. In fact, the wireless feature is one of the significant advantages of the magnetic actuation method, but it also brings challenges to the multifunctional implementation of capsules in the complex GI environment. The specific motion of the capsule in a completely unconstrained state imposes strict requirements on the design and control of the applied magnetic field. Meanwhile, the constraint of the limited medical capsule size restricts the design space of the magnetic source inside the capsule in wireless mode. Therefore, most developed magnetically driven capsules typically suffer from deficiencies such as large size (> size #00 capsules, $8.53 \times 23.3$ mm)[42], low ratio of the volume of loaded drug to the total volume of the capsule (low RDC)[43], strict magnetic field requirements (settled direction or high amplitude more than 50 mT) and simple functions. A detailed list of capsule parameters is available in Supplementary Table S1. It's worth noting that, through innovations in capsule structure and materials, some small capsules or those driven by low magnetic fields have been developed[37,44,45]. However, they typically come at the cost of sacrificing other key characteristics. For instance, the reported small capsules require a high magnetic field (> 100 mT) and have a low RDC (< 0.1)[44]. The capsules driven by low magnetic fields also face issues such as complex structures or single functionality without active control[37,45]. Additionally, significant changes in the external structures of these capsules occur during drug release or sampling process, which could allow considerable resistance to the functionality of the capsules in an unstructured and narrow environment (such as the small intestine).

In this work, we developed multifunctional magnetically driven capsules with distinct design concepts, which we refer to as MagCaps. We combined magnetic soft robotics with capsule design and proposed a controllable magnetic soft valve based on the competitive interactions between the magnetic gradient force and magnetic torque (Fig. 1a). The soft valve is self-closed in the absence of a magnetic field, and it can be opened by applying the magnetic field $\mathbf{B_{ha}}$ (Fig. 1b), which benefits from the dual-layer structure and magnetization design of the magnetic soft valve. Meanwhile, we proposed a multimodal actuation method for decoupling and regulating the overall and local kinetic responses of the MagCaps. The MagCaps can complete the targeted drug delivery (Fig. 1d) and sampling function (Fig. 1e) in the stomach or intestine after entering the human body via oral intake (Fig. 1c). Finally, we developed six MagCaps (Fig. 1f) and explored their magnetodynamic response and potential applications. More detailed descriptions (dimensions and function) of these MagCaps in Fig. 1f are provided in Supplementary Note S1, Table S2, and Fig. S1.

## Results

### Design and performance of magnetic soft valve of MagCaps
Establishing an efficient interactive channel between the interior of the capsule and the external environment (i.e., the gastrointestinal tract) is crucial for achieving functions such as drug release and sampling. To achieve this objective, this work proposes constructing a magnetically controlled valve on the capsule with the following functionalities: 1) Normally closed to prevent leakage without a magnetic field, 2) Ability to open via a magnetic field for controllable interaction with the GI tract, and 3) Allowing controlled movement of the entire capsule under a magnetic field. As magnetic soft composites have excellent performance in terms of deformability, controllability, and flexibility[46–51], we introduce the magnetic soft composites with embedded neodymium-iron-boron (NdFeB) microparticles into the soft valve design. Meanwhile, we develop a dual-layer structure soft valve consisting of a movable magnetic leaf and a fixed magnetic frame (Fig. 1a–i). The magnetization patterns of the magnetic frame and magnetic leaf are set as axial magnetization ($\mathbf{M_1}$) and radial symmetric magnetization ($\mathbf{M_2}$), respectively. This specific magnetization design allows the soft valve to automatically close due to the magnetic attractive force between the leaf and frame in the absence of a magnetic field, ensuring a tight seal for the capsule. Moreover, when a strong enough external magnetic field is applied, the leaf can have an inward U-shape deformation under magnetic torque and open the valve (Fig. 1b-ii). This feature ensures the implementation of the second function mentioned above, and the deformation pattern similar to the in-opening window can avoid the leaf deformation from being affected by the GI environment. It's worth noting that, unlike existing magnetic valves, which use the gradient magnetic force for their opening ways[52,53], the proposed soft valve opens independently of the field gradient of the externally applied magnetic field. This independence is undoubtedly potent for remote control in practical applications, as the magnetic field gradient rapidly diminishes with increased distance between the capsule and the magnetic source. Additionally, the developed soft valve features automatic closure due to its inherent magnetic attractive force, whereas previous valves still required an external gradient magnetic field. Lastly, but equally important, by imparting unidirectional magnetization to the magnetic frame and bidirectional magnetization to the magnetic leaf, the entire capsule exhibits a net magnetization characteristic, laying the groundwork for the controlled overall movement of capsules under the influence of a magnetic field.

The open-closed characteristics of the magnetic soft valve are directly determined by the magnetization strength of the magnetic frame, which is closely associated with the mass fraction of NdFeB particles in its composites (Fig. 2a). Repeated results of the magnetization measurements are shown in Supplementary Fig. S2a. Actually, excessive or insufficient NdFeB particle content is not suitable for practical applications because the former will require a considerably high magnetic field to open the valve, while the latter will cause the

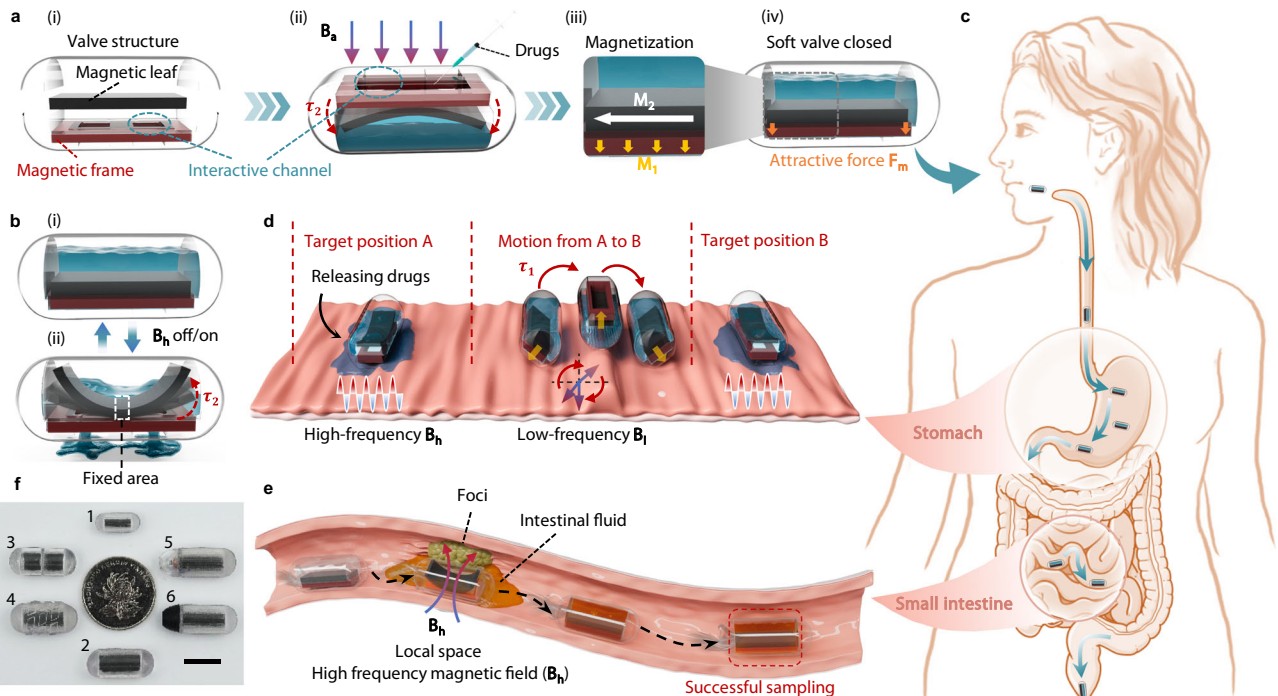

**Fig. 1 | Structural design, function and application of MagCaps. a** Structure of the MagCaps and drug injection process. **b** Schematic diagram showing the opening and closing process of the soft valve under an external magnetic field. **c** Schematic diagram of the human GI tract and the ingestible capsule, the figure was partly generated using Servier Medical Art, provided by Servier, licensed under a Creative Commons Attribution 3.0 unported license[71]. **d** Schematic diagram showing the multi-target drug release process in the stomach. **e** Schematic diagram of the sampling process in the small intestine. **f** Photographs of the developed 6 MagCaps (Miniature Cap1, Visualization Cap2, Dual-module Cap3, Threaded Cap4, Light-assisted Cap5, Thermal-assisted Cap6). Scale bar: 10 mm.

problem of poor capsule sealing. To this end, we explore the effect of the external magnetic field $\mathbf{B_a}$ on the deformation profiles of magnetic leaf, where magnetic frames with different magnetic particle content are considered, as shown in Fig. 2b. In the meanwhile, to facilitate a better understanding of our specific experimental conditions, we have organized the corresponding magnetic field conditions for each experiment in Supplementary Tables S3 and S4.

After calibrating the magnetic field amplitude with a magnetometer (G93, Shenzhen Coliy Technology Development Co., Ltd., China), the critical magnetic fields for opening the valve are measured in these cases, as shown in Fig. 2c. It can be concluded that the NdFeB particle content has a positive relation with the magnetic gradient force between the magnetic frame and leaf, the required magnetic field to open the valve, and the sealing performance of the MagCaps. For instance, at a mass fraction of 30% NdFeB particles, the valve's opening magnetic field indicating the magnitude of the externally applied magnetic field at which the magnetic leaf detaches from the magnetic frame is approximately 6.8 mT. Increasing the mass fraction to 70% results in an opening magnetic field of about 22.6 mT, marking a 232% increase. This demonstrates that the opening magnetic field can be controlled by adjusting the NdFeB particle content in the magnetic frame, offering a means to achieve multilevel valve regulation.

Furthermore, Fig. 2d shows the bending angles of the magnetic leaf under the effect of magnetic frames with different content of NdFeB particles. In the figure, we increase the applied magnetic field from 0 mT to 40 mT and then decrease it to 0 mT. It can be observed that when the magnetic leaf is separated from the magnetic frame, the content of NdFeB particles has basically no impact on the deformation angle of the magnetic leaf (Supplementary Fig. S3). This is because the magnetic gradient force decreases sharply as the distance between the magnetic frame and the leaf increases. This phenomenon suggests that a simplified simulation, which doesn't account for the magnetic

gradient force, is not only applicable for predicting the deformation of magnetic leaf in the absence of a magnetic frame (as depicted in Supplementary Fig. S4) but is also suitable for analyzing the deformation of magnetic leaf once separated from the frame. The details of the simulation model are described in Supplementary Note S2.

A good sealing performance can ensure that there is no drug leakage in the capsule before reaching the target area, and that the sample in the capsule is not contaminated after sampling. However, the unstructured GI environment poses a significant challenge to the sealing performance, as the capsule undergoes a long period of movement during which extrusion and collision could occur. Therefore, it is necessary to evaluate the sealing performance of the capsule. First, we configure a push rod made of polylactic acid (PLA + , Shenzhen Esun Industrial Co., Ltd., China) with a known bottom area (D = 1.5 mm) at the top surface of the precision weigher (AX324ZH, Changzhou OHAUS Instruments Co., Ltd., China) to exert force, propelling the magnetic leaf until it reaches a critical state of detachment from the magnetic frame. The force value at this moment is measured and converted into pressure, as shown in Fig. 2e. It is reported that the maximum pressure of the digestive phase does not exceed 25 mm Hg (3.33 kPa)[54]. Therefore, MagCaps with magnetic frame mass fraction exceeding 50% can maintain a seal under gastrointestinal pressure. Then, in response to potential collision scenarios, the MagCaps containing the Ponceau S (Aladdin Biochemical Technology Co., Ltd., China) solution is placed in a beaker filled with water. The beaker is fixed on a shaker (OS-20, JOANLAB Equipment Co., Ltd., China) to simulate the vigorous peristalsis of the human GI under extreme conditions (Fig. 2f). The shaker speed is set to 100 rpm (Supplementary Fig. S5 and Movie 1), significantly exceeding the maximum frequency of gastrointestinal contractions in normal humans (9–12 contractions per minute[55]. Figure 2g shows that when the mass fraction of NdFeB particles in the magnetic frame is more than 50%, the leakage rate of the capsule is less than 2% (100 rpm for 5 min),

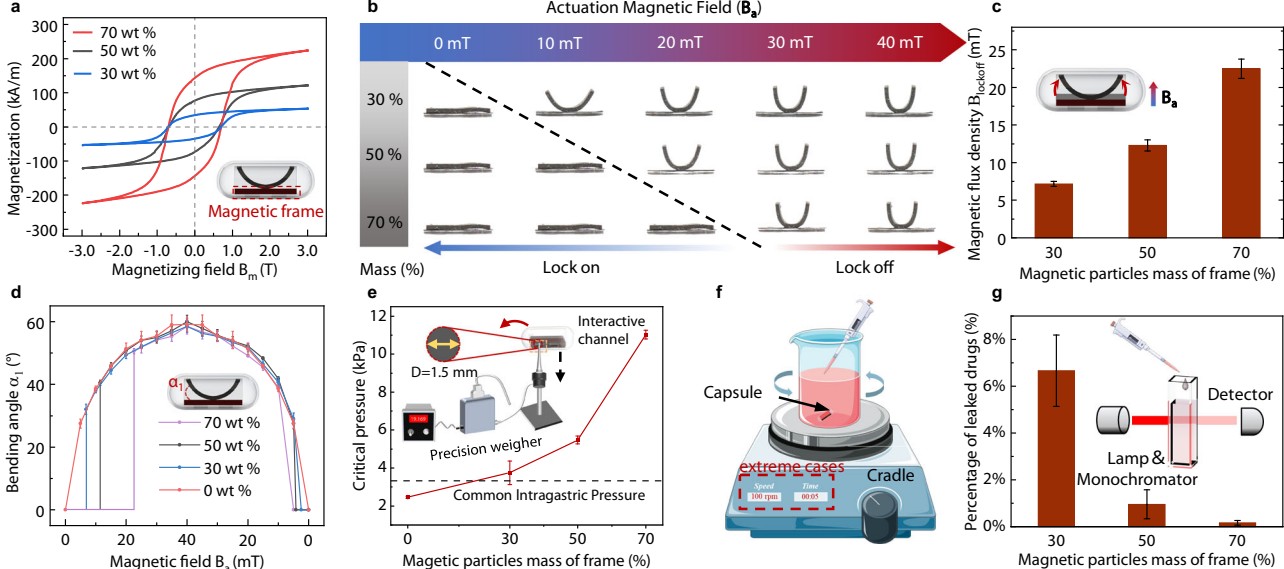

**Fig. 2 | Magnetic characterization and sealing test of MagCaps. a** Magnetic hysteresis loops of the magnetic soft composites containing NdFeB microparticles and PDMS. Source data are provided as a Source Data file. **b** Deformation profiles of the magnetic leaf when the applied magnetic field strength increased from 0 mT to 40 mT. **c** Magnetic fields required to open the magnetic valve for magnetic frames with the magnetic powder content are set to 30%, 50%, and 70% ($n = 3$, data are presented as mean values +/− SD). **d** Measured bending angles with different magnetic powder content under the static magnetic field ($n = 3$, data are presented as mean values +/− SD). **e** Critical pressures required for the magnetic leaf detachment from the magnetic frame with varying magnetic powder mass fractions ($n = 3$, data are presented as mean values +/− SD). **f** Schematic diagram of the sealing test, where the capsules containing the drug were placed on a shaker and tested at the extreme case. **g** Sealing performance of capsules for magnetic frames with different magnetic powder content, which can be revealed by the percentage of leaked drugs ($n = 3$, data are presented as mean values +/− SD). Source data are provided as a Source Data file. The Fig. 2e–g were partly generated using Servier Medical Art, provided by Servier, licensed under a Creative Commons Attribution 3.0 unported license[71].

indicating that the developed capsule has good sealing performance and can meet the practical application requirements.

## Actuation strategy and dynamic behavior of MagCaps

The functional realization of MagCaps is closely related to magnetic responses of the global structure and the local valve. However, the realization of independent and flexible regulation of these two responses is challenging as they share the same internal magnetization characteristics and the external magnetic field. In this section, we propose a dual-frequency magnetic actuation strategy to decouple the global locomotion of the developed MagCaps and the local deformation of the valve, as shown in Fig. 3a. On the one hand, thanks to the specific magnetization structure of the magnetic soft valve, the capsule exhibits a generally single magnetic orientation characteristic overall (Fig. 1a-ii). Therefore, the capsule is capable of moving globally from Pos_1 to Pos_2 by applying a low-frequency magnetic field $B_l$ (1 ~ 5 Hz). This global locomotion behavior is highly related to the magnetic torque $\tau_1$ generated by the low-frequency magnetic field $B_l$ and the residual magnetization $M_1$. On the other hand, the local deformation is controlled by applying a relatively high-frequency magnetic field (20 ~ 60 Hz), which can be attributed to magnetic torque $\tau_2$ produced by the high-frequency magnetic field $B_h$ (20 ~ 60 Hz) and the residual magnetization $M_2$.

We measure the horizontal motion information of the MagCaps under rotating magnetic fields with different frequencies (Supplementary Movie 2), as shown in Fig. 3b. These magnetic fields are generated by a self-developed magnetic field generation system (Supplementary Fig. S6b–d). The capsule demonstrates good controllability and high motion efficiency at frequencies less than 5 Hz, accomplishing a maximum motion rate of about 30 mm per cycle at 1 Hz. The locomotion process of the capsule under a single-cycle magnetic field is depicted in Fig. 3b-i, which includes a magnetic field actuation phase and an inertia phase. When the frequency surpasses

5 Hz, the capsule's rotation shows a significant hysteresis with respect to the magnetic field. Meanwhile, the increase in frequency causes serious instability in the inertial phase of the capsules, which significantly reduces the controllability (Fig. 3b-ii). Most notably, when the frequency is increased to 20 Hz, the capsule produces almost no displacement in 3 cycles (0.15 s), indicating that the low-frequency magnetic field can efficiently and stably drive the global locomotion of the capsule. The main reason for this phenomenon is that the overall rotation of the capsule requires a certain relaxation time.

When the magnetic field frequency exceeds 5 Hz, the capsule cannot complete a full cyclic rotation, decreasing the motion efficiency. Another piece of information gained from this phenomenon is that the anchoring function can be achieved at a high frequency. To verify this point, the anchoring performance of the capsule on a plane with a diameter of 20 cm is tested under a high-frequency magnetic field (Supplementary Movie 3), and the results are shown in Fig. 3c. A relatively large magnetic field (30 mT) is adopted for this test, which considers the actual magnetic field requirements of the open valve. The experiments show that the anchoring performance of the capsule improves with the magnetic field frequency. When the field frequency increases from 10 Hz to 30 Hz, the capsule's away rate decreases from 50 mm/s (2.56 body/s) to 3.08 mm/s (0.157 body/s). This clearly indicates that the capsule achieves a good anchoring performance under a high-frequency magnetic field, even in the smooth test plane, which facilitates precise drug release and sampling for target areas.

The relationship between the local deformation behavior of the soft valve and the magnetic field frequency is analyzed. The results are depicted in Fig. 3d-i, where the capsule stays fixed to exclude the effect of the global locomotion. The experiments demonstrate that when the frequency is less than 30 Hz, the maximum bending angle of the magnetic leaf does not differ significantly. As the field frequency increases, the bending angle of the leaf shows an obvious decreasing trend. For instance, the maximum bending angle decreases by nearly

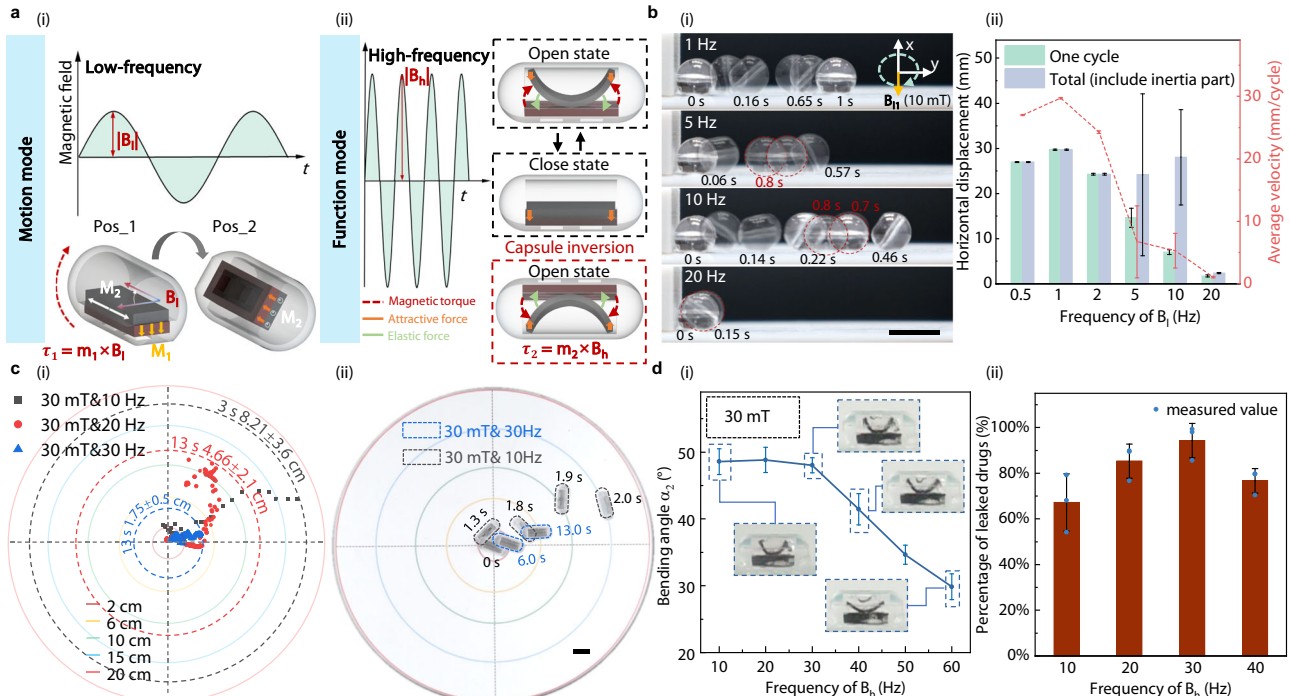

**Fig. 3 | Decoupling principle and performance of the global locomotion and local deformation of MagCaps under magnetic fields. a** Schematic diagram showing the principles of global locomotion (i) and local deformation (ii) under low-frequency (1 - 5 Hz) and high-frequency (20 - 60 Hz) magnetic fields. **b** Global locomotion performance testing of capsules under varying frequency magnetic fields (1 - 20 Hz) generated by the Helmholtz coil system: locomotion process (i) and measured locomotion information (ii) (*n* = 3, data are presented as mean values +/− SD). **c** Anchoring performance assessment of capsules when the magnetic field frequencies range from 10 Hz to 30 Hz: distribution map of landing points (i) and motion process for two cases (ii). **d** Local deformation of magnetic leaves (i) and drug release performance (ii) of capsules under magnetic fields with varying frequencies (10 - 60 Hz) (*n* = 3, data are presented as mean values +/− SD). Source data are provided as a Source Data file. Scale bar: 10 mm.

34% at the frequency of 60 Hz. This is mainly because the leaf deformation requires a certain response time, e.g., it takes only 4 ms for the magnetic field to rise to its maximum value when the field frequency reaches 60 Hz, whereas the response time for leaf deformation should be greater than 8.2 ms.

Furthermore, we measure the drug release ratio of the capsule at different frequencies within 10 s, as shown in Fig. 3d-ii. We use methyl blue dye (Aladdin Biochemical Technology Co., Ltd., China) as the labeled drug and calculate the ratio between the sample group and the control group after drug release to estimate the actual drug release ratio. It can be gathered that with the increase of magnetic field frequency, the drug release rate of the capsules exhibits the change rule of increasing and then decreasing, and finally achieves the highest drug release efficiency at 30 Hz. This is because although a complete magnetic leaf deformation can be achieved at a low magnetic field frequency, the induced low interaction frequency between the valve and the surrounding liquid decreases the drug release efficiency. When the frequency is too high, the decreased deformation amplitude of the magnetic leaf also affects the drug release efficiency. To enhance our understanding of the MagCaps' working principle, we conducted a two-dimensional fluid-solid coupling simulation using COMSOL Multiphysics software to qualitatively analyze the flow field situation (Supplementary Note S3). It is worth noting that if the capsule is not fixed, it can rotate with the changing direction of the magnetic field. This behavior will allow the external magnetic field to be aligned with the magnetization direction of the magnetic frame of the valve, thus inhibiting the valve opening. Fortunately, as aforementioned, with the increase of magnetic field frequency, the capsule has an anchoring function at high frequencies that effectively avoids this problem, which further increases the difference of the leaf deformation at low and high frequencies. These results clearly illustrate that controlling

the magnetic field frequency can independently achieve high-efficiency global locomotion and local deformation of the MagCaps.

## Integrated functionality of targeted delivery, drug release, and sampling

In addition to the traditional image capturing function, the diagnosis and treatment of GI diseases put forward urgent requirements for providing other functions of MagCaps, such as the active targeted delivery in a large spatial area[42], site-directed drug release for reducing drug dosage and preventing drug side effects[56], and sampling for biological information in GI tract[6,7]. Therefore, in this section, we further verify the effectiveness of the proposed capsule for achieving integrated multi-functionality through systematic experimental studies.

Using the electromagnetic actuation system shown in Fig. 4a, we manipulate the motion direction of these MagCaps using a handle controller. Figure 4b clearly shows that the capsule reaches the target position from the starting point through the S-bend track made by PLA. Then, an alternating magnetic field (30 mT, 30 Hz) is applied to the target area to complete the drug release process (Supplementary Movie 4). Detailed modeling and analysis of the kinematic process are presented in Supplementary Note S4. Similarly, Fig. 4c shows the sampling process of the capsule in a U-shaped channel made by PLA. When it reaches the sampling area, an alternating magnetic field is applied to open and close the soft valve repeatedly for sampling. Subsequently, the alternating magnetic field is turned off at the end of the sampling process, finally, the capsule is controlled by a controller to oeuvre it out of the sampling area.

We also conduct experimental validation in an anatomical model to demonstrate the adaptive motion properties of the capsule and its high controllability of drug release in more complex environments.

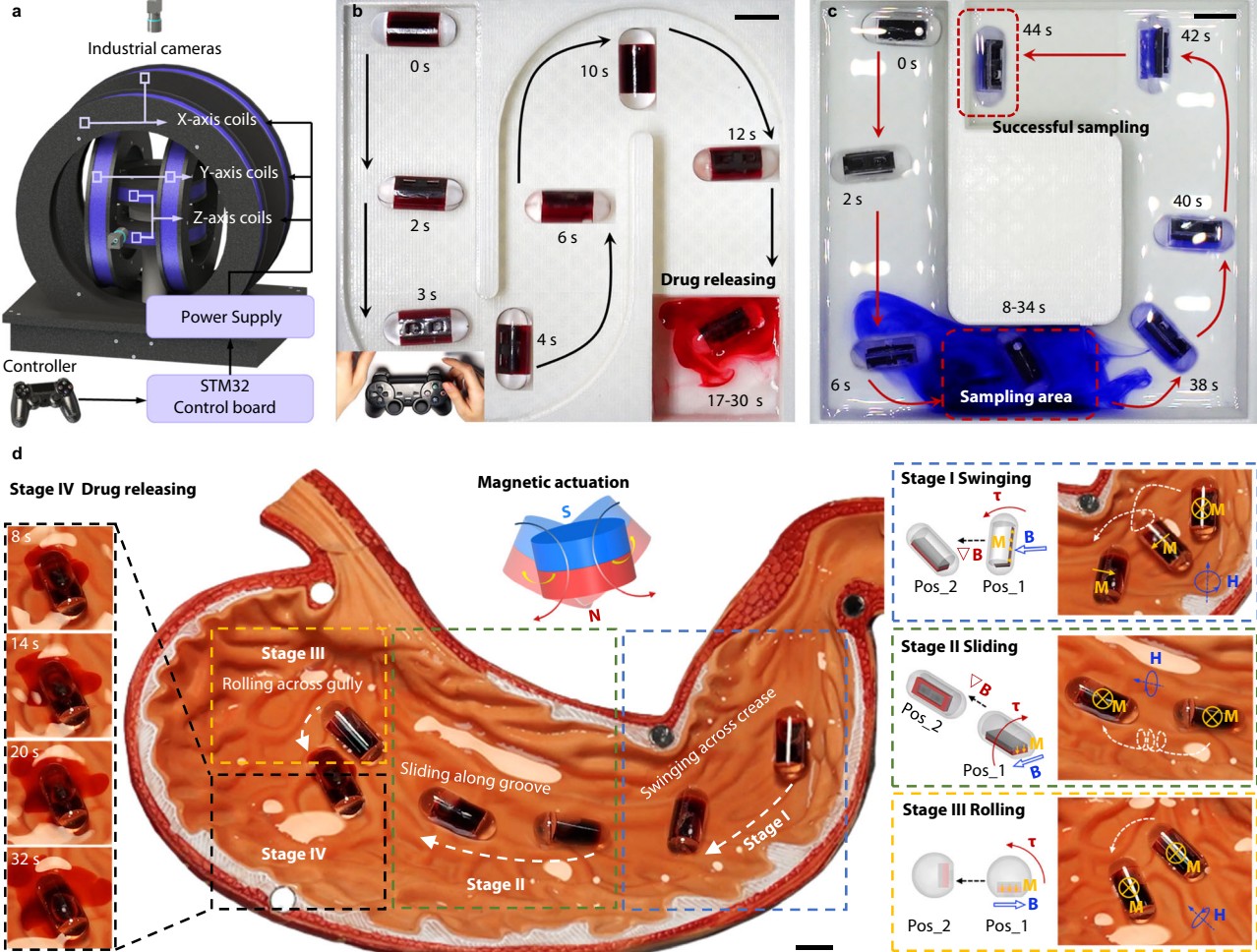

**Fig. 4 | Functional demonstration of MagCaps in targeted drug delivery, release and sampling. a** Schematic diagram of the developed Helmholtz coil actuation system. **b** Targeted drug delivery of MagCaps rolling along an S-shaped testing platform and the final drug release driven by a uniform magnetic field. **c** Targeted sampling of MagCaps in a U-shaped testing platform under a uniform magnetic field. **d** Targeted drug delivery using a permanent magnet (D50 × L20 mm, N35) and drug releasing with a 30 Hz, 30 mT high-frequency magnetic field (generated by the coil in Fig. S6e) in the anatomical model of the stomach, which can be divided into four stages: swinging, sliding, rolling and releasing. Scale bar: 10 mm.

First, the gradient force and magnetic torque generated by a N35-grade permanent magnet with a diameter of 50 mm and height of 30 mm (D50 mm × H30 mm, Beijing Jiuci Technology Co., Ltd., China) are used to control the multimodal motion of the capsule in the anatomical model of the human stomach, such as swinging (Stage I), sliding (Stage II), and rolling (Stage III). Subsequently, a high-frequency magnetic field is applied to complete the drug release once the capsule reaches the designated target area (Stage IV, Supplementary Movie 5), as shown in Fig. 4d. The detailed description of the capsule movement and the drug release process is provided in Supplementary Note S5.

As highly automated equipment can significantly improve safety in the surgical setting, we further develop a magnetic actuation system for multi-target transport and drug delivery of MagCaps in the ex vivo test. This equipment mainly consists of a robotic arm (AUBO-i10, AUBO Robotics Technology Co., Ltd., China, Payload: 10 kg), a permanent magnet (Chengdu Juyu Magnetic Material Co., Ltd., China, D80 mm × H80 mm, N35), a handle controller, and a control panel, as displayed in Fig. 5a. In the ex vivo porcine stomach, the 6-degree-of-freedom robotic arm (Supplementary Fig. S6g) loaded with a permanent magnet is manipulated by a handle controller to move and rotate, so that the capsule can move under the action of magnetic gradient force (Fig. 5b) and magnetic torque (Fig. 5c). When it arrives at the predetermined target positions A and B, respectively, a high frequency

alternating magnetic field generated by a coil is applied so that the drug can be released from the capsule (Supplementary Movie 6), as shown in Fig. 5d.

We further focus on the sampling function of the MagCaps in the ex vivo small intestine tissue. We replace the permanent magnet in Fig. 5a with a poly-magnetic coil (Fig. 5e and Supplementary Fig. S6h) for accurate sampling of local regions, where an iron core with a trapezoidal cross-section is introduced to concentrate the magnetic field around the coil surface. More details can be found in Methods. We establish a 2D axisymmetric coil model to quantify magnetic field distribution and measure field strength near the coil surface, as shown in Fig. 5f, g. Simulated and experimental magnetic field data closely align and display distinctive gradient decay characteristics. For instance, at a 4 V discharge voltage, the magnetic field within 7 mm decreases from over 50 mT to under 30 mT. This gradient decay feature allows accurate sampling. To validate this point, we conduct drug release experiments in the pipe model at different distances, as shown in Fig. 5h. With a 40 Hz sinusoidal voltage at 4 V, there is no leakage of the drug inside the capsules when the distance between the capsule and the coil decreases from 3.6 cm to 1.39 cm, and drug release occurs solely when the capsule is centered within the coil (Supplementary Movie 7). This result suggests that the magnetic field generated by the coil has a relatively high spatial resolution (± 1 cm) and can be utilized

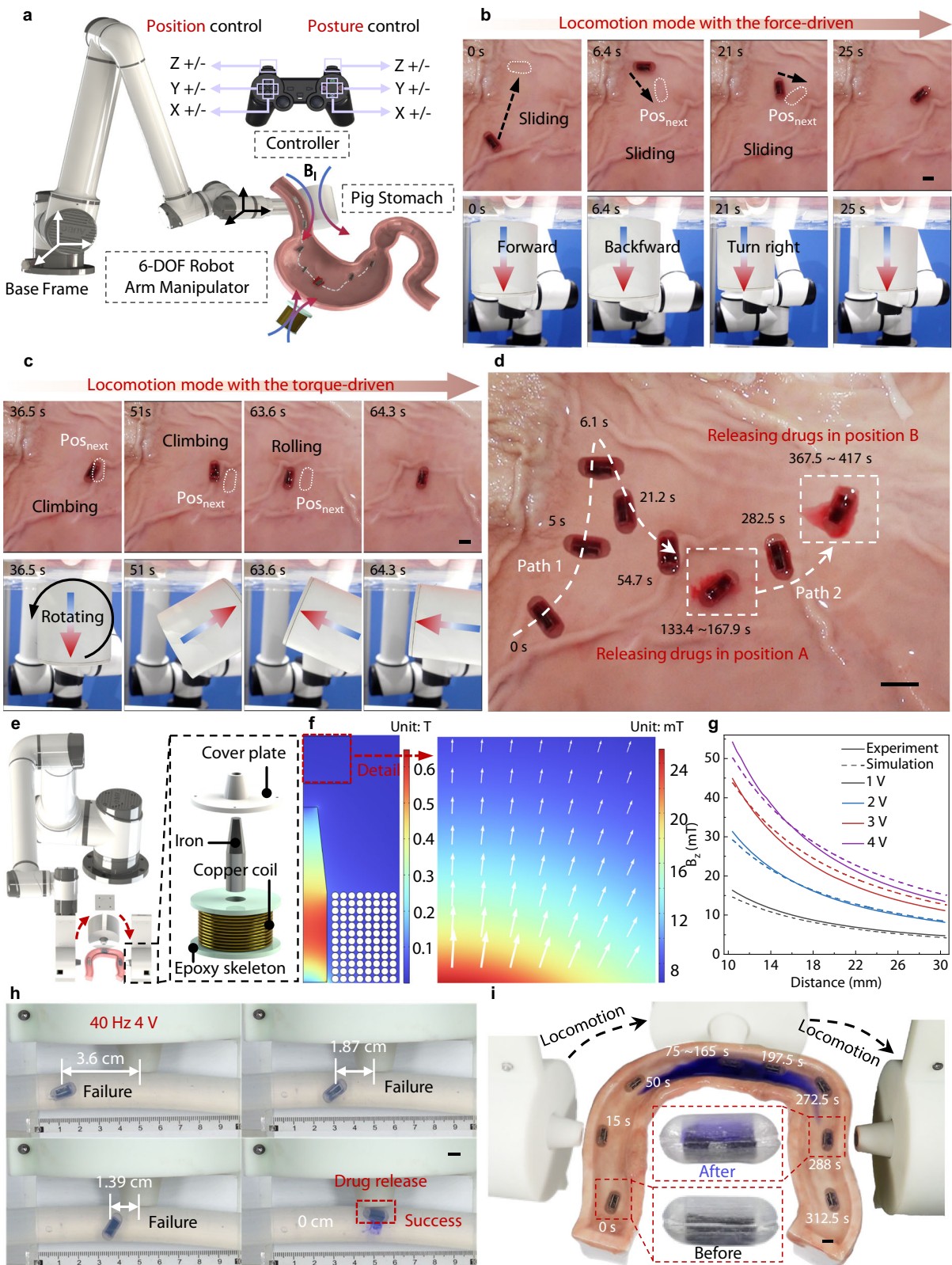

**Fig. 5 | Ex vivo demonstration of multi-target drug release and sampling in porcine stomach and small intestine, respectively. a** Schematic depiction of the employed magnetic actuation system that utilizes a 6-DOF magnetic manipulator and a coil for ex vivo testing within a porcine stomach. **b** Sliding mode of the capsule under the magnetic gradient force. **c** Rolling mode of the capsule under the magnetic torque. **d** Demonstration of multi-targeted drug delivery of the capsule with a high-frequency (30 Hz) magnetic field. **e** Schematic depiction of the employed magnetic actuation system that utilizes a 6-DOF magnetic manipulator

for ex vivo sampling testing and incorporates a coil with a trapezoidal cross-section iron core. **f** Magnetic field distribution map around the coil surface computed by the COMSOL Multiphysics software. **g** Measured and simulated magnetic fields under different coil voltages along the central axis. Source data are provided as a Source Data file. **h** Drug release testing performed at varying distances from the coil center aimed to demonstrate the fixed-point sampling capability of this coil setup. **i** Demonstration of active locomotion and the sampling process within an ex vivo model of the porcine small intestine. Scale bar: 10 mm.

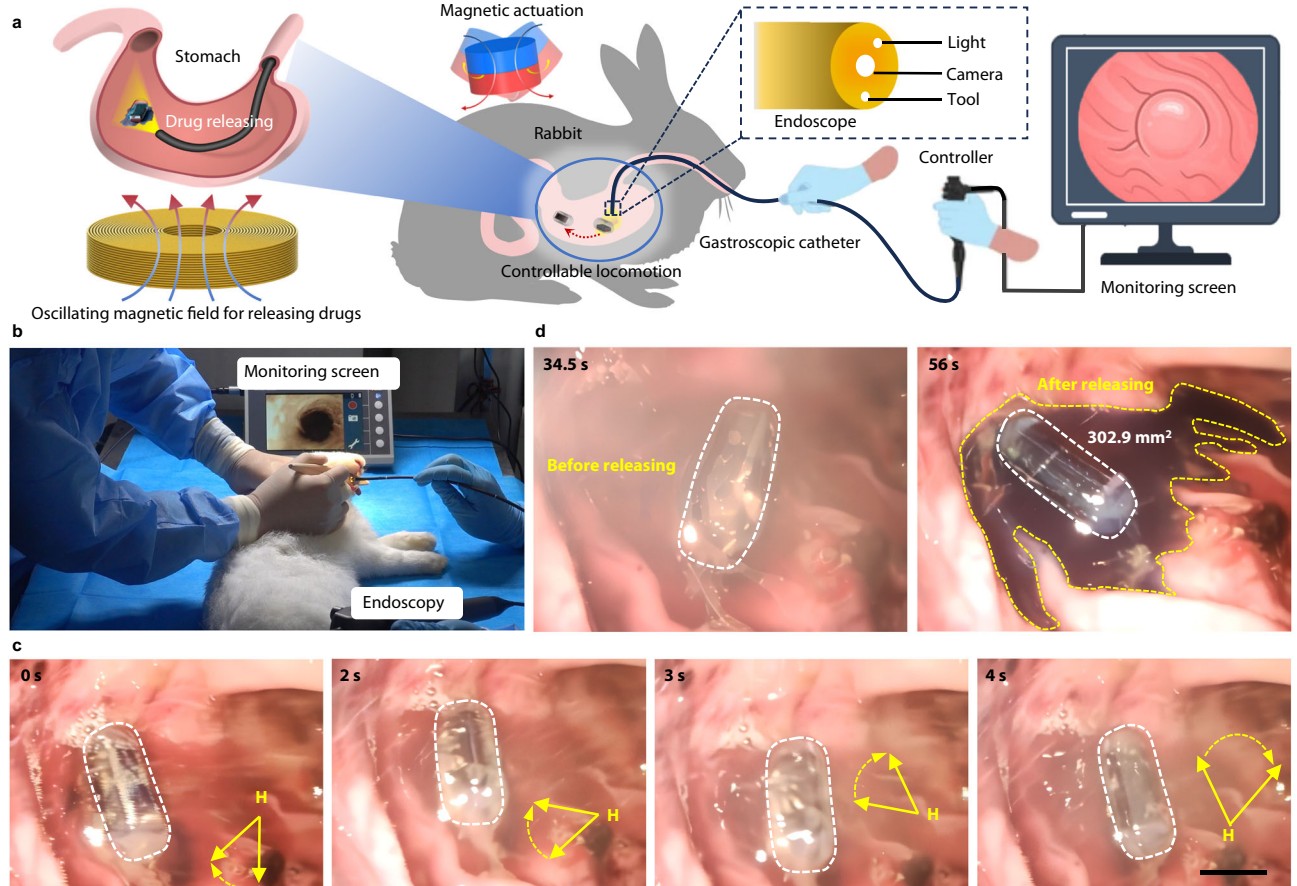

**Fig. 6 | In vivo demonstration of MagCaps in a rabbit's stomach model.**
**a** Schematic diagram of the in vivo testing system. The locomotion and drug release processes are observed using an electronic endoscope. The figure was partly generated using Servier Medical Art, provided by Servier, licensed under a Creative Commons Attribution 3.0 unported license[71]. **b** Experimental site image. **c** Diagram of the targeted motion process of the capsule under an externally applied magnetic field generated by a permanent magnet (D80 × 80 mm, N35), where the distance between the magnet and the rabbit stomach is ~15 cm. **d** Photographs showing the region surrounding the capsule within the rabbit's stomach before and after the drug release. Scale bar: 10 mm.

to achieve drug release or sampling in a specified area, such as different regions of the intestine, with centimeter-level accuracy.

Finally, we perform an experimental validation in an ex vivo model of the small intestine, the procedure of which is shown in Fig. 5i. The coil causes the capsule to move due to the magnetic gradient force that is generated by the current in the DC mode, while the AC mode regulates the state of opening and closing of the magnetic soft valve, which in turn enables sampling (Supplementary Movie 8). The comparison graph of the capsule before and after sampling shows that it changes from the original white color to blue color (Fig. 5i). This clearly demonstrates that the developed MagCaps can effectively perform the liquid collection task in the specified target area, which provides a new technological pathway for tumor screening and microbiota sampling.

To further test the motility of our developed MagCaps in a confined narrow space and verify their feasibility for biomedical applications, we conduct an in vivo drug delivery test in a rabbit's stomach, where similar rabbit models are previously used in drug delivery and release studies within the GI tract[51,57,58]. We use a portable endoscope for simultaneous tracking of the capsule delivery and drug release process (Supplementary Movie 9). Figure 6a shows the overall structure and experimental process of this test. Prior to the start of the experiments, a certain amount of anesthetic (intraperitoneal injection of 3% pentobarbital sodium 3 ml/kg) is injected into the rabbit to alleviate its pain and ensure the safety of the experimenters, as shown in Fig. 6b.

After the rabbit ingests the capsule, we use the endoscope (2.9 mm CMOS electronic endoscope; Karl Storz, Inc., Germany) to enter its stomach along the esophagus to observe the motion trajectory of the capsule. The directional movement of the capsule is carried out in the rabbit's stomach using a permanent magnet (Chengdu Juyu Magnetic Material Co., Ltd., China, D80 mm × H80 mm, N35), where it can achieve flexible directional rolling under the traction of a rotating magnetic field within 0–4 s, as shown in Fig. 6c. Subsequently, a high-frequency magnetic field is applied to the body surface projection of the stomach through an electromagnetic coil (Fig. S6e) placed below the rabbit at 34.5 s, which allows the liquid exchange between the drug inside the capsule and the gastric juice. A total of 22 s after the magnetic field application, the methyl blue solution stain (labeled drug) inside the capsule is released (Fig. 6d), where the area of the drug release is only about 302.9 mm². These animal experiments demonstrated that the MagCaps proposed in this work are feasible and effective for targeted drug delivery in living animals.

## Expansion of capsule functionality
We further demonstrate its four extended functions by designing the local structure of the MagCaps and widening the frequency of the applied magnetic field. These functions include dual-drug release, mucus removal, wireless light and heating.

Achieving a multi-drug release, especially a selective release, is highly challenging as it imposes strict demands on the active control of

the capsules[59]. By adjusting the content of NdFeB particles in the magnetic frames of the valve, we develop a dual-module configuration for MagCaps to achieve the controlled release of two types of drugs. As Fig. 7a shows, the left and right parts of the magnetic frames are composed of magnetic soft polymers with a mass fraction (NdFeB: PDMS) of 50% and 70%, respectively. The required magnetic fields for opening the valve on both sides are not the same due to the difference in the attractive gradient magnetic force. When an external magnetic field is applied at a lower level ($|\mathbf{B_{h1}}|$ is about 15 mT), the dual-module capsule operates in Mode I (left-side open), while it switches to Mode II (both sides open) when the magnetic field increases to $\mathbf{B_{h2}}$ ($|\mathbf{B_{h2}}|$ is about 25 mT), and the driven magnetic field is generated by the coil in Fig. S6e powered by an AC supply (SP300VAC5000W, Beijing Quantian Technology Co., Ltd., China).

To validate the effectiveness of the MagCaps (Supplementary Movie 10), we inject Methyl Blue and Ponceau S into the left and right compartments of the capsule, respectively (Fig. 7b). The Methyl Blue starts to leak significantly from the left side of the capsule after 6 s of applying the alternating magnetic field ($\mathbf{B_{h1}}$). It is almost completely released in 22 s, whereas there is no leakage of the Ponceau S from the right side of the capsule during this process. When the alternating magnetic field increases to 25 mT ($\mathbf{B_{h2}}$) at 23 s, the Ponceau S begins to be released. As the concentration of the solution inside the capsule decreases, it can be clearly observed that the magnetic leaves on both sides of the valve are bent at 30 s, which is also consistent with our expected design. When the alternating magnetic field is turned off after the complete release of the Ponceau S (38 s), the color of the solution inside the capsule turns transparent, which fully proves that the Methyl Blue and Ponceau S have been completely released. In the meantime, the dual-module capsule can also release both drugs directly under the high magnetic field $\mathbf{B_{h2}}$ (Supplementary Fig. S7). Based on these results, it can be gathered that the dual-module MagCaps have the advantages of strong controllability and flexible drug release, where drugs can be released sequentially and simultaneously.

Oral drug delivery of large molecules is constrained by the degradation environment and malabsorption in the GI tract, e.g.,

peptides such as insulin and vancomycin have an oral bioavailability of less than 1%[60]. To solve this problem, Srinivasan et al. incorporated threaded features onto the outer surface of the capsule and introduced a motor-drive rotation to locally clear the mucus layer[20]. This design innovation offers an effective mechanism for the enhancement of drug absorption. Inspired by this, we implement similar functionality by developing mucus-clearing MagCaps with a threaded shell. It is noted that the rotation of the capsule reported by Srinivasan et al.[20] is triggered in response to the pH change, necessitating no additional manipulation, thus constituting its most notable advantage, which shows great convenience and implementation in future applications. However, subject to its trigger mode, it is difficult to achieve the action at a specific site or a specific time. In contrast, the magnetically actuated method needs to rely on an external magnetic field system but can bring about a substantial enhancement of controllability.

In our application, the rotation of the capsule accelerates the diffusion of the drug in the region by applying a medium-frequency magnetic field (5–10 Hz) after the completion of drug release (Supplementary Movie 11), as shown in Fig. 7c and Supplementary Fig. S8. Meanwhile, the interaction between the threaded shell of the capsule and the folds of the small intestine can remove the mucus from the inner wall of the small intestine and allow release or sampling, where the latter facilitates the collection of associated tissues such as mucus hairs, and thus improves the sampled information (Fig. 7d–i).

The threaded capsule is placed on the surface of the small intestinal tissue, and a medium-frequency (10 Hz, 15 mT) magnetic field is applied to rotate it for 10 min. Subsequently, the absorbance of the collected intestinal fluid is assessed using a UV-vis spectrophotometer (UV-2600i, SHIMADZU Co., Ltd.), as shown in Fig. 7d-ii. In multiple replicated experiments, the absorbance of the solution is $0.61 \pm 0.53$ after undergoing the smooth capsule treatment. Compared with the former, the absorbance of the solution with the threaded capsule treatment increases by 239.3% (absorption peak at 290 nm), reaching a value of $2.07 \pm 0.24$. The increased absorbance of the liquid samples signifies that the interaction between the threaded capsule and the fold of the small intestine breaks down the barrier that

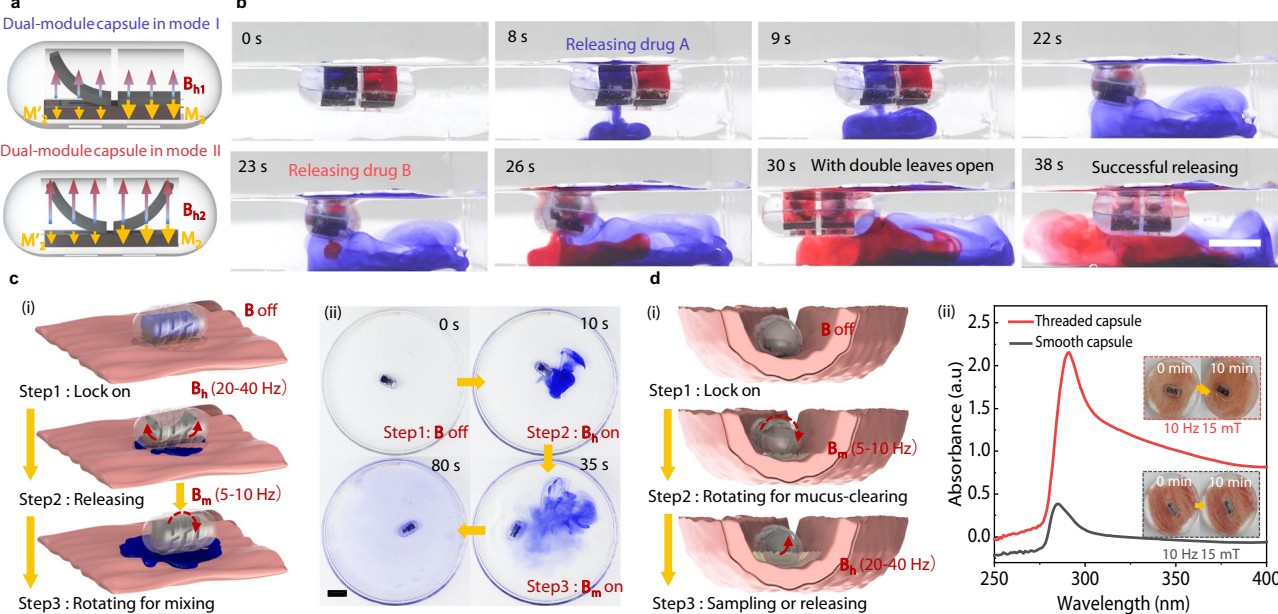

**Fig. 7 | Structure extension design and functional demonstration of MagCaps. a** Schematic diagram of leaf deformation in a dual-module capsule. The magnetic particles' mass fraction of the magnetic frame in the left module is 50%, and it is 70% in the right module. **b** Sequential and controllable release processes of drugs A and B in the dual-module capsule. The open magnetic field of the left module is about 15 mT, and that of the right module is about 25 mT. **c** Magnetic capsule structure with a helical groove and its application for the drug mixing process. **d** Schematic diagram of mucus-clearing (i) and the comparative absorbance (250–400 nm) of the two capsules after 10 min of rotation in the lining of the small intestine (ii). Source data are provided as a Source Data file. Scale bar: 10 mm.

prevents the drug from reaching the epithelial layer. This in turn enhances the absorption rate of the drug in the small intestine. This optimized capsule design enhances oral drug absorption through local drug delivery, increased drug dispersion and mucociliary clearance, which provides a potential pathway for high-efficiency oral drug delivery of large molecules.

In vivo light/thermal-assisted therapy is a novel therapeutic method for inducing apoptosis or necrosis in tumors and other cells[61,62]. By combining low-frequency (<1 Hz) and ultra-high-frequency (UHF, 20 - 80 kHz) magnetic fields, magnetic soft robots are able to deform and move while providing electrical or thermal energy, thereby offering the possibility to achieve more functions[63,64]. Here, we further propose a multi-frequency regulation method based on low-high-ultra frequency magnetic fields for MagCaps. In the function of wireless light therapy, an LED induction light is integrated into the end of a capsule to provide an additional magneto-electric function, as shown in Fig. 8a-iii. When there is no UHF magnetic field, no current is generated in the induction coil and the LED light is in the off state. In the presence of the UHF magnetic field (35.7 kHz), a change in the magnetic flux in the induction coil generates induced eddy currents, which in turn light up the LED, as shown in Fig. 8a-ii. The decoupling of motion and magneto-electric function is achieved using the low-frequency and UHF magnetic fields to control the capsule motion and the LED light switch, respectively, and the four letters "HUST" are successively lit on the prescribed path, as shown in Fig. 8b (Supplementary Movie 12). This extended design supports the expansion of MagCaps into wireless optical therapy.

Furthermore, we develop MagCaps with an additional magneto-thermal function, as depicted in Fig. 8c-i. The magneto-thermal part of the capsule consists of a mixture of $Fe_3O_4$ nanoparticles (50–300 nm,

Aladdin Biochemical Technology Co., Ltd., China) and silica gel elastomer Ecoflex 00-10 (Beijing Angelcrete Art Landscaping Co., Ltd., China), and the heating processes for three mass fractions of 10%, 30% and 50% under a UHF magnetic field (38.5 kHz) generated by the induction heating machine (45 kW, Guangdong Taiguan Power Technology Co., Ltd., China) are shown in Fig. 8c-ii. It is shown that the higher the content of $Fe_3O_4$ particles, the higher the rate of heating, e.g., when the mass fraction is 50%, the highest temperature reached after heating for 60 s is equal to 98 °C by an infrared temperature imager (Ti400 + , Fluke Shanghai Co., Ltd., China). On this basis, the multi-frequency regulation method is applied to achieve the integrated control of the capsule's target delivery, fixed-point drug release, and wireless magneto-thermal functions (Supplementary Movie 13), as photographed in Fig. 8d. The capsule is moved to a specified position under a low-frequency magnetic field (Fig. 8d-i). It then performs drug release under a high-frequency magnetic field (Fig. 8d-ii) and is heated by a UHF coil (38.5 kHz) for 60 s (Fig. 8d-iii), reaching a max temperature of 110 °C (Fig. 8d-v). Finally, the capsule is moved away from the target area with a low-frequency magnetic field (Fig. 8d-iv), resulting in the drug being heated to 39.15 °C (Fig. 8d-vi). These results clearly demonstrate the integrated control of drug transport, controlled release, and on-demand heating. In summary, the decoupled modulation and functionally integrated design of the multiple magnetic field frequency response of MagCaps provide the foundation for their applications in light/thermal-assisted therapy for GI diseases.

## Discussion

In this work, we introduced, magnetic soft robotics into the capsule design for multifunctional MagCaps, breaking away from the

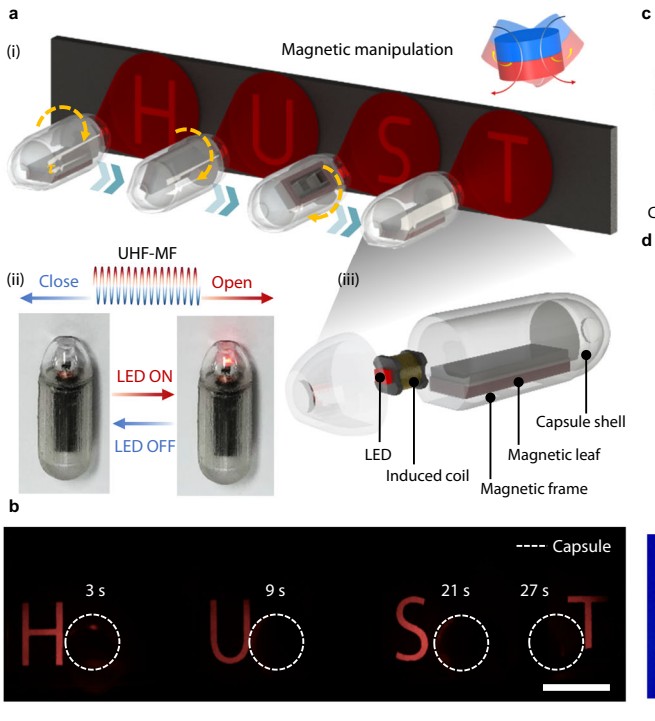

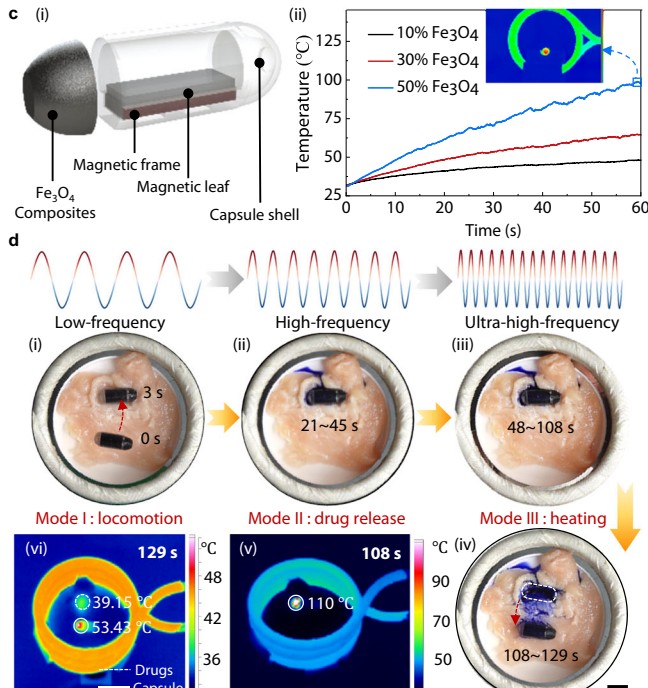

**Fig. 8 | Magnetic excitation extension design and functional demonstration of MagCaps. a** Light-assisted capsule with additional wireless luminescence functionality: schematic diagram of function integration of targeted locomotion and luminescence (i), structural diagram of the capsule (ii) and magnetic field-induced LED lighting (iii). **b** Experimental demonstration of targeted locomotion driven by a low-frequency (<1 Hz) magnetic field (generated by a permanent magnet D50 × L20 mm, N35) and luminescence supplying with ultra-high-frequency (35.7 kHz) magnetic field. **c** Structural diagram of a thermal-assisted capsule with additional

wireless heating functionality (i) and heating process for varying magnetic nanoparticle content (ii). (n = 3). Source data are provided as a Source Data file. **d** Multifunctional demonstration of the magnetic-thermal capsule: targeted locomotion under a low-frequency (<1 Hz) field generated by a magnet (i), drug release under a 30 Hz, 30 mT high-frequency field (ii), heating under an ultra-high-frequency (38.5 kHz) magnetic field (iii), locomotion away from the target region under a low-frequency (<1 Hz) field (iv), and temperature of the capsule and the target region at different times (v-vi). Scale bar: 10 mm.

conventional design mode of built-in permanent magnets. Specifically, by constructing a magnetic soft valve in the capsule with a double-layer ferromagnetic composite film, we developed an effective means for bidirectional interactions between the capsule cavity and the GI environment, which provided a pivotal solution for the integrated drug release and sampling functionalities of capsules. Meanwhile, we put forward a multi-frequency response-based actuation strategy for MagCaps with decoupling control of global and local magneto-dynamic responses. This strategy overcame the bottlenecks of traditional capsules, such as single-motion mode, limited functions, and poor extendibility, and provided key technology support for achieving the active targeted transport and local multi-mode magnetic response. These achievements have empowered the developed MagCaps to accomplish essential functions effectively across a range of applications, such as targeted drug delivery and sampling, dual-drug release, mucus clearance, and wireless light and heating. This multifaceted medical functionality provides the foundation for the wide application of the developed MagCaps in the diagnosis and treatment of GI diseases.

We clearly validated multiple potential application functions of the developed capsules; however, future work should combine specific pathology experiments to optimize the capsule structure design and evaluate the advantages of these MagCaps. For instance, compared with traditional methods, quantitative analysis with respect to reducing the drug dosage, the medication frequency, and the side effects due to the use of locally targeted drug release should be discussed; It is necessary to determine the optimal size of capsules based on the required drug release or sampling volume in practical application scenarios; It is also interesting to leverage the sampling function of MagCaps to investigate the correlation between intestinal flora and diseases across different intestinal regions. Simultaneously, it is worth noting that large animal models like pigs, with GI environments very similar to humans, should be prioritized in future studies.

Additionally, regarding further improvements in capsule technology or performance, the following aspects are worth considering. Firstly, concerning the overall motion of MagCaps, as observed from Fig. 5, it is evident that the combined application of magnetic torque and gradient magnetic force enriches their overall motion patterns. However, predicting and precisely controlling the capsule's motion under the simultaneous influence of these two driving forces still requires further in-depth research. Secondly, the integration of additional actuation sources, such as built-in motor[20] or mechanical springs[60], into capsule design could complement magnetic actuation strategies, aiming to simplify the external magnetic sources and actuation strategies. Thirdly, the emergence of capsule endoscopy has enabled comprehensive visualization of the entire GI tract, showing promising potential as an effective clinical tool for monitoring gut health[6,65] Therefore, the integration of the imaging module from existing capsule endoscopy into the MagCaps could significantly enhance real-time monitoring capabilities during capsule operation. Lastly, it is favorable to know the position and orientation of the capsule because it can guide the application of magnetic field for the desired motion, a key factor for this is the introduction of real-time localization techniques[66]. Considering that medical CT imaging methods usually exist in radiation for patients and health-care workers, we can account for some assisted positioning technologies for the long-term dynamic positioning of MagCaps in the GI environment, such as ultrasound[67] and magnetic[68,69] assisted localization technology.

## Methods

### Preparation of magnetic soft valve

The main preparation process for the magnetic leaf and magnetic frame involves stirring, injection molding and heat demolding, laser cutting, and magnetization, as detailed in Supplementary Fig. S9.

The magnetic leaf is composed of silicone elastomer (Ecoflex 00–10) and NdFeB microparticles with an average size of 5 μm (MQP-15-7, Tianjin Magnequench Co., Ltd., China). In the experiments, these materials were sufficiently mixed in a container at a 1:1 mass ratio. Subsequently, the blend was mixed in a planetary mixer at 2000 r.p.m. for 60 s, followed by defoaming at 2200 r.p.m. for 45 s. Then, the homogeneous magnetic slurry was poured into a mold with a rectangular groove made of polytetrafluoroethylene and cured at room temperature for four hours. Finally, the prepared magnetic soft material was symmetrically folded and subjected to a pulsed magnetic field applied in the vertical direction for magnetization.

The magnetic frame is composed of polydimethylsiloxane (PDMS, SYLGARD 184, Dow Corning Co., USA) and NdFeB microparticles. In the experiments, the magnetic powder and PDMS were mixed in different mass ratios (7:3, 1:1, 3:7). Subsequently, the curing agent was added, and the mixture was thoroughly defoamed. The thoroughly blended magnetic mixture was then poured into the prepared mold and heated in an oven at 100 °C for 1 h. Finally, the prepared soft material for the magnetic frame was subjected to a pulsed magnetic field applied in the vertical direction for magnetization.

It is noted that the rationale behind using different elastic materials for these two components can be explained as follows: Ecoflex 00-10, with its lower Young's modulus, allows for greater deformability, facilitating larger deformations of the magnetic leaf. Conversely, to enhance the magnetic attractive force between the magnetic leaf and frame, a higher magnetic powder content is desirable for the magnetic frame. To address this requirement, the excellent flowability of PDMS proves advantageous in injection molding, enabling the attainment of a smooth and flat surface on the magnetic frame.

### Fabrication of MagCaps

The capsule structure of the developed MagCaps consists of a magnetic leaf (black), a magnetic frame (red) and a capsule shell (transparent), as shown in Fig. 1a. If drug injection into the capsule is required, it can be achieved by inducing deformation of the magnetic leaf through the application of an external magnetic field (Fig. 1a-ii). For different experimental scenarios, two different types of capsule shells have been manufactured using UV-curable 3D printing technology, including transparent shells (Shenzhen Chenno 3D Technology Co., Ltd., China) and MED610 shells (Guangzhou Chengxing Digital Technology Co., Ltd., China). The former is beneficial for observing the deformation of the magnetic leaf and its interaction with the magnetic frame in dynamic studies. The latter is a PolyJet photopolymer manufactured by Stratasys Ltd., which is more biocompatible[70] and can ensure that the capsule is not corroded by gastric acid and other substances in biological validation experiments. For detailed information, please refer to Supplementary Fig. S10, which demonstrates that the capsule does not soften when immersed in acid. These two types of capsules were manufactured identically, except for the differences in material and size (Supplementary Fig. S1). Specifically, the capsule shell was processed by splitting it into upper and lower parts to facilitate the formation of the magnetic leaf, the magnetic frame and the capsule shell. The magnetic frame after magnetization was first embedded in the groove of the lower part of the capsule shell. Subsequently, the center of the magnetic leaf was fixed to the central area of the lower part with silicone adhesives (Sil-Poxy, Beijing Angelcrete Art Landscaping Co., Ltd., China). Finally, the upper and lower parts of the capsule shell were glued together using a medical-grade adhesive (Loctite 435, Henkel Investment Co., Ltd., China).

### Magnetic characterization

The magnetization of samples (magnetic composite materials) was measured with a superconducting quantum interference device (MPMS3-SQUID, Quantum Design Co., Ltd., USA) using the vibrating sample magnetometer option. The maximum symmetrical magnetic

field was varied from −30000 Oe to 30000 Oe in a step size of 200 Oe. Under the same magnetic field condition, we set up three samples (converted to volume by measuring mass with a densitometer) to obtain the hysteresis line and avoid measurement errors.

## Sealing-ability test method

Firstly, in Supplementary Note S6, we explored the linear relationship between the concentration of Ponceau S and its absorbance (Supplementary Fig. S11a, b). During the test, the experimental group placed a capsule containing a known mass ($m_1$) of Ponceau S (3 mg/ml) in a water tank containing a known mass ($m_2$) of water. The capsule was fixed on a shaking table set at parameters of 100 rpm for 5 min. Simultaneously, a control group was established: the solution of Ponceau S with mass $m_1$ was directly poured into the water with mass $m_2$ and thoroughly stirred, representing 100% drug leakage. Each case was repeated three times, and the absorbance at the corresponding wavelengths of both the experimental control groups was measured using a UV-vis spectrometer (UV-2600i, SHIMADZU Co., Ltd.). The ratio of the absorbance of the experimental group to that of the control group can be considered as the proportion of the leaked capsule drug.

## Drug release rate test

In the drug release measurement, we used Methyl blue solution as the labeling drug. Similar to the sealing-ability testing, we measured the absorbance of different concentrations of Methyl blue solution before the experiment, using a UV-vis spectrophotometer, and plotted the standard curve of Methyl blue concentration versus absorbance (Supplementary Note S6 and Fig. S11c–d). We injected a 3 mg/mL Methyl blue solution (3 mL) into a 40 mL volumetric flask containing distilled water and measured its absorbance as the reference value. Subsequently, we injected the same concentration and volume of Methyl blue solution into the capsules and placed them in 40 mL of distilled water for drug delivery experiments at different frequencies. The absorbance ratio of the solution after release to the reference value was measured as the drug release ratio. This allows us to measure the drug release proportions from the capsules at different magnetic field frequencies (Fig. 3d-ii). In addition, the drug release time of the capsule was controlled by the timer built into a controller. The timer was set to turn on when the digital-to-analog converter (DAC) started to output a sine wave. The output waveform was stopped when the timer count overflows. The controller was based on an STM32F407GTZ chip (STMicroelectronics, Supplementary Fig. S6d), and the DAC module was based on a DAC8563 chip (Texas Instruments) and a power amplifier (Fig. S6f, Nanjing Foneng Technology Industry Co., Ltd).

## Ex vivo experiment setup

Ideal model validation: In the experiments presented in Fig. 4a–c, Supplementary Movies 1 and 3, a 3D Helmholtz coil system (Supplementary Fig. S6b, YP Magnetic Technology Development Co., Ltd., China) was employed to move the capsule in S-curves and U-shaped grooves. Real-time motion control of the capsule in the curved grove was performed using the wireless controller, which transmitted the position of the joystick in real time to the microcontroller via Bluetooth signals. The microcontroller calculated the corresponding spatially rotating magnetic field waveform, comprising x-, y-, and z-axis sine waves with different phases, thus dynamically altering the 3D magnetic field in space in real time. Handheld permanent magnets produced a magnetic field in arbitrary 3D directions for the capsule in the anatomical model (Fig. 4d). The deformation and motion of the capsule were captured using a video camera (RX100 VI, Sony).

Animal experiments ex vivo: The drug release equipment was displayed in Fig. 5a. It mainly included a 6-DOF robotic arm, a control panel, a wired controller, a permanent magnet for driving the capsule, and a copper coil for releasing drugs. The copper coil (Fig. S6e) for generating the alternating magnetic field was fabricated by Yichang Zhuokai Technology Co., Ltd., China. The photograph in Fig. 5e displays the poly-magnetic coil used in the sampling process, comprising a resin cover plate, an iron core, and a copper coil. In addition, an AC power supply (SP300VAC5000W) manufactured by Beijing Quantian Technology Co., Ltd. was adopted to supply sinusoidal current to the coil with an iron core.

## In vivo animal experiment

This research aims to collect basic data about the targeted drug delivery and drug release function of the proposed MagCaps in living animals. All animal experiments were conducted in accordance with the protocol approved by the Institutional Animal Care and Use Committee of Huazhong University of Science and Technology (IACUC number: 3381). Four male New Zealand rabbits weighing 2.5–3 kg (purchased from WQJX-BIO Technology, Co., Ltd., China; License No. SCXK 2018-0020) were utilized for endoscopically monitored capsule testing experiments. Food intake, such as feed and hay for each rabbit, was restricted two days before the experiment, and both were in a fasting state for 24 h before the endoscopic operation. The rabbits were fed the magnetic capsule within 1 h before the experiment. Before the experiment began, the rabbits were anesthetized by an intraperitoneal injection of 3% pentobarbital sodium (3 ml/kg). In the experiment, an electronic endoscope (2.9 mm CMOS electronic endoscope; Karl Storz, Inc., Germany) was first applied to observe the situation in the gastric cavity of the rabbits. The capsule was positioned and subsequently, the endoscope was utilized to properly clean up the remaining stomach contents. To simulate and clearly present the drug release process of the capsule, a biological stain (methyl blue, 2 ml: 20 mg, Jumpcan Pharmaceutical Group Co., Ltd. China) served as the drug in this work. In the drug delivery experiment, real-time high-definition images from the endoscopy were used to monitor and observe the drug release process under magnetic actuation.

## Wireless light and heating experiment setup

In the wireless light experiment, we used a permanent magnet (50 mm × 30 mm, N35) and a homemade coil (Fig. S6i) to generate a low-frequency (<1 Hz) and ultra-high-frequency (37.5 kHz) magnetic field, respectively. In the wireless heating experiment, the magneto-thermal part of the capsule was heated by an ultra-high-frequency magnetic field (38.5 kHz) generated by an induction heating machine (45 kW, produced by Guangdong Taiguan Power Technology Co., Ltd., China), as shown in Fig. 8c-ii. Meanwhile, the temperature of the area heated by the ultra-high-frequency magnetic field was measured by an infrared temperature imager (Ti400+, Fluke Shanghai Co., Ltd., China).

## Statistical analysis

All quantitative experimental values are presented as mean ± standard deviation (SD) of the mean.

## Reporting summary

Further information on research design is available in the Nature Portfolio Reporting Summary linked to this article.

# Data availability

All data supporting the findings of this study are available within the article and its supplementary files. Any additional requests for information can be directed to, and will be fulfilled by, the corresponding authors. Source data are provided with this paper.

# Code availability

All codes for simulation are available upon request.

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

## Acknowledgements

The work was financially supported by National Science Foundation of China (51821005). E.S. acknowledges support from the Discipline Crossover Foundation of Wuhan National High Magnetic Field Center (WHMFC202127). We would like to thank Xiaohan Wu for her assistance in drawing, and Limeng Du for proofreading this article.

## Author contributions

Conceptualization: Q.C., E.S., L.L.; Methodology: Q.C., Y.S., W.Z.; Investigation: Y.S., W.Z., J.G., L.X., X.Z., H.W., S.O., Y.C, R.L., J.L., Z.J., D.C.; Visualization: Y.S., W.Z., L.X.; Funding acquisition: L.L., X.H., Q.C., E.S., Y.L.; Project administration: Q.C., L.L., X.H., K.C., Y.L.; Supervision: Q.C., L.L.; Writing—original draft: Y.S., W.Z., Q.C., J.G.; Writing—review and editing: Q.C., Y.S., W.Z., J.G., W.Q., L.X., X.Z., H.W., S.O. All authors contributed to the discussion.

## Competing interests

The authors declare no competing interests.
