## [Peer Review File · Nature Communications]

REVIEWER COMMENTS

Reviewer #1 (Remarks to the Author):

The authors developed small-scale magnetic capsule robots which have multiple potential applications including targeted drug delivery. The work is novel and the results are comprehensive. The results also support the conclusions well. However, the description of the design of the magnetic capsule robots is very brief. The methodology is not thorough as well. It is hard for readers to understand the design and movement mechanism etc. The procedures cannot allow other researchers to replicate or confirm the work. These should be addressed in the revised version. The rest are of high quality.

Reviewer #2 (Remarks to the Author):

The work presents details on a soft magnetic particle/elastomer composite membrane and frame integrated within an additively manufactured capsule. The soft magnetic membrane/frame structure acts as a valve that can be actuated at low magnetic frequencies, which does not interfere with the active magnetic propulsion of the capsule itself at high frequencies. Similar work on the use of magnetic valves and magnetic actuation has been done elsewhere for gastrointestinal clinical applications by people such as Eric Diller, Pietro Valdastri and Metin Sitti. I was surprised such work was not referenced in the text. The capsule was characterised and utilised in several applications, ranging from passive drug delivery where by adjusting the degree by which the membrane/valve opened the diffusion of drugs can be adjusted and mucus clearing using a threaded capsule. I was particularly surprised the mucus-clearing application did not cite Traverso's recent work in this field.

There are numerous spelling and grammar issues, for example, *in-vitro/in-vivo* etc. are typically italicised, that need to be addressed throughout but my main concern with this work is the lack of detail in the main text which would impede the reproduction of the work. This includes the dimensions of the capsule, dimensions of the membrane, provenance of many of the materials used such as the NdFeB particles, the PDMS, the Ecoflex, PLA materials, the SQUID devices, The Fe₃O₄ particles, the magnetometer, UV-VIS spectrometer, lisinopril and so on. Additionally, the results are often described in vague terms which impairs the readers' ability to judge the significance of the results, for example "This characteristic makes it easier to open the valve over long distances, as generating a sufficiently strong magnetic gradient is often challenging" - define long, define strong. Define what the low, medium and high frequencies each time for clarity. "This result suggests that the magnetic field generated by the coil has a high spatial resolution and can be utilized to achieve drug release or sampling in a specified area, e.g., different regions of the intestine, with a centimeter accuracy." - what is considered high spatial resolution in this context, how many centimeters, is it +/- 1cm, or +/- 10cm. "A certain amount of

anesthetic is injected into the rabbit prior to the start of the experiments to alleviate its pain and ensure the safety of the experimenters," - how much is a certain amount? There are numerous similar examples such as this dotted throughout the text that need to be addressed. From a methodology perspective, some clarification on why the authors switched from using ex-vivo porcine tissue to an in-vivo small animal model (rabbit) would be welcome. A rabbit is not the typical animal model used for GI devices, usually a porcine model is used due to superficial similarities with the human GI tract.

Reviewer #3 (Remarks to the Author):

The authors were able to generate a capsule for drug delivery, sampling, and locomotion using a novel magnetic leaf valve design. For additional functionality, onboard lights, heat generating materials, and mucus clearing shell were incorporated to the compact device. By adjusting the frequency of an external rotating magnetic field either above or below the capsule's step-out frequency, sampling/drug delivery and locomotion were independently addressed.

While the capsule's multifunctionality was clearly demonstrated, more discussion on valve modeling, locomotion modeling, and limitations to their capsule and locomotion strategy could have been provided. Moreover, different magnetic actuation systems were used for different tests (i.e., Helmholtz coil systems, permanent magnets, and coil system on robot arm etc.). It was unclear at times which actuation system was being used and how capsule and valve behaviour measured with the 3D coil system at the beginning of the paper would change with the use of permanent magnets and 2D coil at the end of the paper.

The following are some comments and suggestions to further clarify their work.

Introduction

- The authors identified capsule size as one of the limitations in the introduction (line 78 – 83). Capsules with smaller sizes than capsule 00 (D 8.53 mm x L23.3 mm) have been previously reported with potential for both drug delivery, sampling, and locomotion.

i.P. Shokrollahi et al., Blindly Controlled Magnetically Actuated Capsule for Noninvasive Sampling of the Gastrointestinal Microbiome, IEEE/ASME Transactions on Mechatronics, 26, 5, 2616-2628, 2021.

ii. A. Abramson, et al. An ingestible self-orienting system for oral delivery of macromolecules. Science, 363, 611–303, 2019.

iii.Jiachen Zhang et al.,Voxelated three-dimensional miniature magnetic soft machines via multimaterial heterogeneous assembly.Sci. Robot.6,eabf0112,2021.

- Moreover, size directly correlates with the volume of drug delivered or sample retrieved. What is the average drug volume or sample volume required for your applications?
- For magnetically actuated capsules, capsule size would also affect the magnetic torque available for drug delivery and locomotion. What would be the smallest capsule size possible that would still ensure successful and sufficient drug delivery with your current magnetic actuation systems?
- What would be classified as high magnetic fields or gradients? Would the authors be able to provide a quantitative number?

Materials, Methods, and Experimental Design

- For magnetic characterization using the SQUID, what were the samples you were measuring? Was it the NdFeB particles or composite material?
- For the capsule sealing tests, why was 100 rpm chosen? Does it correlate with the abdominal pressures or forces in the digestive tract?
- In the in-vivo studies, a permanent magnet was used to control the capsule which would have both magnetic gradient and field. How would you account for field gradients that may affect the precise control of the capsule?
- What type of polymer material is used in MED610?
- What was the size and grade of the permanent magnet in Fig. 4d? Would you be able to provide more details on the various coil systems and robot arm used?
- The dual-mode sampling, light and heating capsule experiments appears to be missing in the methods section.

Results & Discussion

- Fig. 1a (i) and Fig. 1a (ii) – It was not clear where the drug exits the capsule, perhaps the author can label the capsule opening.
- Fig. 1a – It was unclear how the magnetic leaf generates a ‘U’ shape under an external magnetic torque. Authors should show that the middle of the leaf fixed to the center to enable consistent generation of the ‘U’ shape.
- Fig. 1f – The figure label should describe what capsules 1 – 6 are and the scale bar should be labelled.
- Line 128 – 129, “It can be concluded that the NdFeB particle content is positively correlated with the magnetic gradient force...” The sentence should be rephrased as positive correlation sounds like a regression analysis was performed.
- What is the definition of the valve’s opening magnetic field? Is there a specific opening angle this corresponds to?

- Fig. 2a – Magnetic moment usually has units of $A\ m^2$. Do you mean magnetization?
- Fig. 2e – Is α_2 the same angle as α_1 in Fig. 2d?
- Fig. 2d and Fig. 2e seems a bit redundant as it is showing similar trends of bending angle vs. magnetic field.
- Fig. 3d – What is α_3 ? Is it the same angle as α_1 and α_2 ? What are the dots in Fig. 3d(ii)?
- Is there a resonant frequency at which your valves operate at?
- Movie S2 – were the capsule actuated by a continuous rotating magnetic field for the same duration? Why does the last frame in the movie look different from Fig. 3b (i.e., the capsule's distances at 5 Hz and 10 Hz in the video vs. Fig. 3b)?
- The mucus clearing design looks very similar to the RoboCap design by Srinivasan, S. S. et al. RoboCap: Robotic mucus-clearing capsule for enhanced drug delivery in the gastrointestinal tract. *Sci. Robot.* 7, eabp9066 (2022) which you cited. What are the differences between the designs?
- Line 342 – 343 – "...the absorbance in solution increases by 239.3% (absorption peak at 290 nm) and 318.5% (absorption peak at 330 nm)..." Why did you choose to compare absorbance at 290 nm and 330 nm?
- Capsule localization was very briefly discussed and perhaps you can cite some relevant papers.
- How would you turn the capsule around the bend during locomotion? What fields in the xyz directions were applied?
- Movie S9 – The capsule appeared to be stuck in mucus. Would higher frequency have helped move the capsule more effectively? What are your frequency limits before drug delivery is activated?
- How well does the capsule perform under external forces from the body?
- While multifunctionality is an interesting topic for wireless capsules, what are the author's thoughts of having one capsule with many functions versus many capsules with specialized functions? Would this depend on application?

REVIEWER COMMENTS

Reviewer #1 (Remarks to the Author):

The authors developed small-scale magnetic capsule robots which have multiple potential applications including targeted drug delivery. The work is novel and the results are comprehensive. The results also support the conclusions well. However, the description of the design of the magnetic capsule robots is very brief. The methodology is not thorough as well. It is hard for readers to understand the design and movement mechanism etc. The procedures cannot allow other researchers to replicate or confirm the work. These should be addressed in the revised version. The rest are of high quality.

Reviewer #2 (Remarks to the Author):

The work presents details on a soft magnetic particle/elastomer composite membrane and frame integrated within an additively manufactured capsule. The soft magnetic membrane/frame structure acts as a valve that can be actuated at low magnetic frequencies, which does not interfere with the active magnetic propulsion of the capsule itself at high frequencies. Similar work on the use of magnetic valves and magnetic actuation has been done elsewhere for gastrointestinal clinical applications by people such as Eric Diller, Pietro Valdastri and Metin Sitti. I was surprised such work was not referenced in the text. The capsule was characterised and utilised in several applications, ranging from passive drug delivery where by adjusting the degree by which the membrane/valve opened the diffusion of drugs can be adjusted and mucus clearing using a threaded capsule. I was particularly surprised the mucus-clearing application did not cite Traverso's recent work in this field.

There are numerous spelling and grammar issues, for example, *in-vitro/in-vivo* etc. are typically italicised, that need to be addressed throughout but my main concern with this work is the lack of detail in the main text which would impede the reproduction of the work. This includes the dimensions of the capsule, dimensions of the membrane, provenance of many of the materials used such as the NdFeB particles, the PDMS, the Ecoflex, PLA materials, the SQUID devices, The Fe₃O₄ particles, the magnetometer, UV-VIS spectrometer, lisinopril and so on. Additionally, the results are often described in vague terms which impairs the readers' ability to judge the significance of the results, for example "This characteristic makes it easier to open the

valve over long distances, as generating a sufficiently strong magnetic gradient is often challenging" - define long, define strong. Define what the low, medium and high frequencies each time for clarity. "This result suggests that the magnetic field generated by the coil has a high spatial resolution and can be utilized to achieve drug release or sampling in a specified area, e.g., different regions of the intestine, with a centimeter accuracy." - what is considered high spatial resolution in this context, how many centimeters, is it +/- 1cm, or +/- 10cm. "A certain amount of anesthetic is injected into the rabbit prior to the start of the experiments to alleviate its pain and ensure the safety of the experimenters," - how much is a certain amount? There are numerous similar examples such as this dotted throughout the text that need to be addressed. From a methodology perspective, some clarification on why the authors switched from using ex-vivo porcine tissue to an in-vivo small animal model (rabbit) would be welcome. A rabbit is not the typical animal model used for GI devices, usually a porcine model is used due to superficial similarities with the human GI tract.

Reviewer #3 (Remarks to the Author):

The authors were able to generate a capsule for drug delivery, sampling, and locomotion using a novel magnetic leaf valve design. For additional functionality, onboard lights, heat generating materials, and mucus clearing shell were incorporated to the compact device. By adjusting the frequency of an external rotating magnetic field either above or below the capsule's step-out frequency, sampling/drug delivery and locomotion were independently addressed.

While the capsule's multifunctionality was clearly demonstrated, more discussion on valve modeling, locomotion modeling, and limitations to their capsule and locomotion strategy could have been provided. Moreover, different magnetic actuation systems were used for different tests (i.e., Helmholtz coil systems, permanent magnets, and coil system on robot arm etc.). It was unclear at times which actuation system was being used and how capsule and valve behaviour measured with the 3D coil system at the beginning of the paper would change with the use of permanent magnets and 2D coil at the end of the paper.

The following are some comments and suggestions to further clarify their work.

Introduction

- The authors identified capsule size as one of the limitations in the introduction (line 78 – 83). Capsules with smaller sizes than capsule 00 (D 8.53 mm x L23.3 mm) have been previously reported with potential for both drug delivery, sampling, and locomotion.

i.P. Shokrollahi et al., Blindly Controlled Magnetically Actuated Capsule for Noninvasive Sampling of the Gastrointestinal Microbiome, IEEE/ASME Transactions on Mechatronics, 26, 5, 2616-2628, 2021.

ii. A. Abramson, et al. An ingestible self-orienting system for oral delivery of macromolecules. Science, 363, 611–303, 2019.

iii.Jiachen Zhang et al.,Voxelated three-dimensional miniature magnetic soft machines via multimaterial heterogeneous assembly.Sci. Robot.6,eabf0112,2021.

- Moreover, size directly correlates with the volume of drug delivered or sample retrieved. What is the average drug volume or sample volume required for your applications?

- For magnetically actuated capsules, capsule size would also affect the magnetic torque available for drug delivery and locomotion. What would be the smallest capsule size possible that would still ensure successful and sufficient drug delivery with your current magnetic actuation systems?

- What would be classified as high magnetic fields or gradients? Would the authors be able to provide a quantitative number?

Materials, Methods, and Experimental Design

- For magnetic characterization using the SQUID, what were the samples you were measuring? Was it the NdFeB particles or composite material?

- For the capsule sealing tests, why was 100 rpm chosen? Does it correlate with the abdominal pressures or forces in the digestive tract?

- In the in-vivo studies, a permanent magnet was used to control the capsule which would have both magnetic gradient and field. How would you account for field

gradients that may affect the precise control of the capsule?

- What type of polymer material is used in MED610?
- What was the size and grade of the permanent magnet in Fig. 4d? Would you be able to provide more details on the various coil systems and robot arm used?
- The dual-mode sampling, light and heating capsule experiments appears to be missing in the methods section.

Results & Discussion

- Fig. 1a (i) and Fig. 1a (ii) – It was not clear where the drug exits the capsule, perhaps the author can label the capsule opening.
- Fig. 1a – It was unclear how the magnetic leaf generates a ‘U’ shape under an external magnetic torque. Authors should show that the middle of the leaf fixed to the center to enable consistent generation of the ‘U’ shape.
- Fig. 1f – The figure label should describe what capsules 1 – 6 are and the scale bar should be labelled.
- Line 128 – 129, “It can be concluded that the NdFeB particle content is positively correlated with the magnetic gradient force...” The sentence should be rephrased as positive correlation sounds like a regression analysis was performed.
- What is the definition of the valve’s opening magnetic field? Is there a specific opening angle this corresponds to?
- Fig. 2a – Magnetic moment usually has units of $A \cdot m^2$. Do you mean magnetization?
- Fig. 2e – Is α_2 the same angle as α_1 in Fig. 2d?
- Fig. 2d and Fig. 2e seems a bit redundant as it is showing similar trends of bending angle vs. magnetic field.

- Fig. 3d – What is α_3 ? Is it the same angle as α_1 and α_2 ? What are the dots in Fig. 3d(ii)?
- Is there a resonant frequency at which your valves operate at?
- Movie S2 – were the capsule actuated by a continuous rotating magnetic field for the same duration? Why does the last frame in the movie look different from Fig. 3b (i.e., the capsule's distances at 5 Hz and 10 Hz in the video vs. Fig. 3b)?
- The mucus clearing design looks very similar to the RoboCap design by Srinivasan, S. S. et al. RoboCap: Robotic mucus-clearing capsule for enhanced drug delivery in the gastrointestinal tract. *Sci. Robot.* 7, eabp9066 (2022) which you cited. What are the differences between the designs?
- Line 342 – 343 – “...the absorbance in solution increases by 239.3% (absorption peak at 290 nm) and 318.5% (absorption peak at 330 nm)...” Why did you choose to compare absorbance at 290 nm and 330 nm?
- Capsule localization was very briefly discussed and perhaps you can cite some relevant papers.
- How would you turn the capsule around the bend during locomotion? What fields in the xyz directions were applied?
- Movie S9 – The capsule appeared to be stuck in mucus. Would higher frequency have helped move the capsule more effectively? What are your frequency limits before drug delivery is activated?
- How well does the capsule perform under external forces from the body?
- While multifunctionality is an interesting topic for wireless capsules, what are the author's thoughts of having one capsule with many functions versus many capsules with specialized functions? Would this depend on application?

Responses to Comments on “NCOMMS-23-49636”

Dear reviewers:

We express our sincere gratitude for your careful review of our manuscript. Those comments are all valuable and very helpful for revising and improving our paper, as well as the important guiding significance to our researches. According to all comments, the manuscript has been revised carefully. We believe that the modifications have contributed to a significant improvement of our manuscript, making it more suitable for publication in Nature Communications.

The main corrections within the paper have been highlighted using red text, and the responses to the comments are list within the subsequent text sections.

Sincerely Yours,

Prof. Quanliang Cao & Prof. Liang Li

Wuhan National High Magnetic Field Center, Huazhong University of Science and Technology

Response to the Reviewer #1:

General comment: The authors developed small-scale magnetic capsule robots which have multiple potential applications including targeted drug delivery. The work is novel and the results are comprehensive. The results also support the conclusions well. However, the description of the design of the magnetic capsule robots is very brief. The methodology is not thorough as well. It is hard for readers to understand the design and movement mechanism etc. The procedures cannot allow other researchers to replicate or confirm the work. These should be addressed in the revised version. The rest are of high quality.

Response: Thank you for the positive feedback and suggestion. To facilitate a better understanding of the capsule's design and movement mechanism, we made substantive modifications in the following aspects.

(1) **We have rewritten the section titled “Overall structure and principle of the magnetic soft valve”, and provided a comprehensive description of the design of the magnetic soft valve, aiming to enhance its comprehensibility.** Specifically, beginning with an introduction of the valve's functions, we provided a detailed account linking the functionalities to the corresponding proposed structure and magnetization pattern of the valve, thus presenting a better correlation between the functions and the design aspects.

(2) **We have added more discussion on the modeling and motion behavior of the developed capsule.** For instance, a fluid-solid coupling modeling was newly provided to understand the dynamic interaction of internal and external fluids within the capsule induced by the dynamic deformation of the soft valve under alternating magnetic fields. Additionally, we have described how the capsule achieves overall rotation and directional movement in a magnetic field. Please refer to Supplementary Text S3 and S3 for the specific modifications.

(3) **We have provided more detailed structural parameters of the developed capsules,** encompassing both the capsule body and valve parameters (see Supplementary Fig. S1).

(4) **The magnetic field parameters and types of magnetic actuation systems used**

in different application scenarios, are presented in the newly added tables (see Table S3). This presentation enhances the transparency of our work.

We believe that as a result of the aforementioned modifications, the design principle and motion mechanism of the developed capsules will become clearer. Moreover, providing more detailed data will also assist readers in replicating our work. **The main modifications are as follows.**

On page 5 and 6, we made revisions to the text:

“Overall structure and principle of magnetic soft valve. Building an efficient interactive channel between the interior of the capsule and the external environment (i.e., the gastrointestinal tract) is crucial for achieving functions such as drug release and sampling. To achieve this objective, this work proposes constructing a magnetically controlled valve on the capsule with the following functionalities: 1) Normally closed to prevent leakage without a magnetic field, 2) Ability to open via a magnetic field for controllable interaction with the GI tract, and 3) Allowing controlled movement of the entire capsule under a magnetic field. As magnetic soft composites have excellent performance in terms of deformability, controllability, and flexibility⁴⁶⁻⁵¹, we introduce the magnetic soft composites with embedded neodymium-iron-boron (NdFeB) microparticles into the soft valve design. Meanwhile, inspired by household inward-opening windows, we developed a soft valve with a dual-layer structure consisting of a movable magnetic leaf and a fixed magnetic frame (Fig. 1a). The magnetization patterns of the magnetic frame and magnetic leaf are set as axial magnetization (\mathbf{M}_1) and radial symmetric magnetization (\mathbf{M}_2), respectively. This specific magnetization design allows the soft valve automatically closes due to magnetic attractive forces between the leaf and frame in the absence of magnetic field, ensuring a tight seal for the capsule. Moreover, when a strong enough external magnetic field is applied, the leaf can have a U-shape deformation under magnetic torque and open the valve (Fig. 1b). This feature ensures the implementation of the second function mentioned above, and the deformation pattern similar to the in-opening window can avoid the leaf deformation from being affected by the GI environment. It's worth noting that, unlike existing magnetic valves which use gradient magnetic forces for their opening ways^{52, 53}, the proposed soft valve opens independently of the field gradient of the externally applied magnetic field. This

independence is undoubtedly potent for remote control in practical applications, as the magnetic field gradient rapidly diminishes with increased distance between the capsule and the magnetic source. Additionally, the developed soft valve features automatic closure due to its inherent magnetic attractive force, whereas previous valves still required an external gradient magnetic field. Lastly, but equally important, by imparting unidirectional magnetization to the magnetic frame and bidirectional magnetization to the magnetic leaf, the entire capsule exhibits a net magnetization characteristic, laying the groundwork for controlled overall movement of capsules under the influence of a magnetic field.”

Revised Fig. S1, newly added Fig. S13, Fig. S14 and Table S3 in Supplementary Materials are provided as follows.

Fig. S1. Dimension diagram of the MagCaps and valves (Capsule 1-6). All units are in millimeters.

Fig. S13. Fluid exchanging processes within a capsule under the action of an applied magnetic field.

Fig. S14. Multimodal locomotion mechanisms of capsules. (a) Illustration of two typical locomotion modes of Magcaps. (b) Left view of the rolling mode. (c) Top view of the turning mode.

Table S3. Magnetic actuation systems used in different experimental scenarios.

Figure number	Type of magnetic actuation systems	Magnetic field type	Magnetic field amplitude	Magnetic field frequency
Fig. 2b-d	Biaxial Helmholtz coil	Stable	0-40 mT	-
Fig. 3b	Triaxial Helmholtz coil	Rotating (X-Y)	10 mT	0.5-20 Hz

Fig. 3c, d	Self-made coil	Sinusoidal	30 mT	10-30 Hz
Fig. 4b, c	1. Triaxial Helmholtz coil	Rotating	1. 10 mT	1. 1Hz
	2. self-made coil	(X-Y-Z)	2. 30 mT	2. 30 Hz
Fig. 4d	1. Permanent magnet (D50×L20 mm, N35)	1. Rotating	1. -	1. <1 Hz
	2. Self-made coil	2. Sinusoidal	2. 30 mT	2. 30 Hz
Fig. 5a-c	1. A permanent magnet (D80 × L80 mm, N35) on a 6-DOF arm	1. Rotating	1. -	1. <1 Hz
	2. Self-made coil	2. Sinusoidal	2. 30 mT	2. 30 Hz
Fig. 5e-i	A poly-magnetic coil on a 6-DOF arm	Sinusoidal	30 mT	30 Hz
Fig. 6	1. Permanent magnet (D80×L80 mm, N35)	1. Rotating	1. -	1. <1 Hz
	2. Self-made coil	2. Sinusoidal	2. 30 mT	2. 30 Hz
Fig. 7a, b	Self-made coil	Sinusoidal	15 and 25 mT	30 Hz
Fig. 7c, d	Self-made coil	Sinusoidal	15 mT and 30 mT	5-10 Hz and 20-40 Hz
Fig. 8a, b	1. Permanent magnet (D50×L20 mm, N35)	1. Rotating	1. -	1. <1 Hz
	2. Wireless Powered Transmitter Coil	2. Sinusoidal	2. -	2. 35.7 kHz
Fig. 8c, d	1. Permanent magnet (D50×L20 mm, N35)	1. Rotating	1. -	1. <1 Hz
	2. Self-made coil	2. Sinusoidal	2. 30 mT	2. 30 Hz
	3. Wireless heating coils	3. Sinusoidal	3. -	2. 35.7 kHz

Response to the reviewer #2:

General Comment: The work presents details on a soft magnetic particle/elastomer composite membrane and frame integrated within an additively manufactured capsule. The soft magnetic membrane/frame structure acts as a valve that can be actuated at low magnetic frequencies, which does not interfere with the active magnetic propulsion of the capsule itself at high frequencies.

Response: We greatly appreciate your general comment and will provide point-by-point response to the specific comments in the following part.

Comment 1: Similar work on the use of magnetic valves and magnetic actuation has been done elsewhere for gastrointestinal clinical applications by people such as Eric Diller, Pietro Valdastrri and Metin Sitti. I was surprised such work was not referenced in the text.

Response 1: Thank you for highlighting this significant point. We acknowledge that researchers like Eric Diller (Refs. [44, 45]), Pietro Valdastrri (Ref. [34, 35]), and Metin Sitti (Ref. [31-33, 36, 43]) have conducted important relevant studies on this topic. We apologize for not referencing these works in the text. In the revised manuscript, we have included citations to these previous studies and ensure a comprehensive discussion in our manuscript, emphasizing the distinctions between their work and our research in the *Introduction Section*. Meanwhile, we also revised **Supplementary Table S1** to provide a more comprehensive comparison between our work and other existing studies, where the values of RDC and deformation pattern of capsules are newly added for comparison. The main modifications are as follows.

On page 4, we made revisions to the text:

“Therefore, the most of developed magnetically driven capsules typically suffer from deficiencies such as large size (>size #00 capsules, 8.53×23.3 mm)⁴¹, low ratio of the volume of loaded drug to the total volume of the capsule (low RDC)⁴², strict magnetic field requirements (settled direction or high amplitude more than 50 mT) and simple functions. A detailed list of capsule parameters is available in Supplementary Table S1. It's worth noting that, through innovations in capsule structure and materials, some extremely small capsules or those driven by low magnetic fields have been

developed⁴³⁻⁴⁵. However, they typically come at the cost of sacrificing other key characteristics. For instance, the reported small capsules require a high magnetic field (> 100 mT) and have a low RDC (<0.1)⁴³. The capsules driven by low magnetic fields also face issues such as complex structures or single functionality without active control^{44, 45}. Additionally, significant changes in the external structures of these capsules occur during drug release or sampling process, which could allow considerable resistance to the functionality of the capsules in an unstructured and narrow environment (such as the small intestine).”

31. Son, D., Gilbert, H. & Sitti, M. Magnetically Actuated Soft Capsule Endoscope for Fine-Needle Biopsy. *Soft Robotics* **7**, 10–21 (2020).

32. Munoz, F., Alici, G., Zhou, H., Li, W. & Sitti, M. Analysis of Magnetic Interaction in Remotely Controlled Magnetic Devices and its Application to a Capsule Robot for Drug Delivery. *IEEE/ASME Trans. Mechatron.* **23**, 298–310 (2018).

33. Yim, S., Goyal, K. & Sitti, M. Magnetically Actuated Soft Capsule With the Multimodal Drug Release Function. *IEEE/ASME Trans. Mechatron.* **18**, 1413–1418 (2013).

34. Simi, M., Gerboni, G., Menciassi, A. & Valdastrì, P. Magnetic Torsion Spring Mechanism for a Wireless Biopsy Capsule. *Journal of Medical Devices* **7**, 041009 (2013).

35. Beccani, M. et al. Component based design of a drug delivery capsule robot. *Sensors and Actuators A: Physical* **245**, 180–188 (2016).

36. Yim, S. & Sitti, M. Shape-Programmable Soft Capsule Robots for Semi-Implantable Drug Delivery. *IEEE Transactions on Robotics* **28**, 1198–1202 (2012).

41. Chu, J. N. & Traverso, G. Foundations of gastrointestinal-based drug delivery and future developments. *Nat Rev Gastroenterol Hepatol* **19**, 219–238 (2022).

42. Zheng, L., Guo, S. & Kawanishi, M. Magnetically Controlled Multifunctional Capsule Robot for Dual-Drug Delivery. *IEEE Systems Journal* **16**, 6413–6424 (2022).

43. Zhang, J. et al. Voxellated three-dimensional miniature magnetic soft machines via multimaterial heterogeneous assembly. *Sci. Robot.* **6**, eabf0112 (2021).

44. Shokrollahi, P. et al. Blindly Controlled Magnetically Actuated Capsule for Noninvasive Sampling of the Gastrointestinal Microbiome. *IEEE/ASME Trans. Mechatron.* **26**, 2616–2628 (2021).

45. Nam, J., Lai, Y. P., Gauthier, L., Jang, G. & Diller, E. Resonance-based design of wireless magnetic capsule for effective sampling of microbiome in gastrointestinal tract. *Sensors and Actuators A: Physical* **342**, 113654 (2022).

Comment 2: The capsule was characterised and utilised in several applications, ranging from passive drug delivery where by adjusting the degree by which the membrane/valve opened the diffusion of drugs can be adjusted and mucus clearing using a threaded capsule. I was particularly surprised the mucus-clearing application did not cite Traverso's recent work in this field.

Response 2: Thank you for your comment. While Traverso's recent studies^{20, 58} were cited in our previous manuscript, they have indeed not been elaborated in detail. Your reminder is very reasonable. In fact, our exploration of this function expansion of the proposed capsules was inspired by their work. As a result, **in the revised manuscript, we have emphasized this point, provided an overview of their work, and performed a comparative analysis.** This enhancement aims to improve the understanding of the strengths and weaknesses inherent in both approaches. The main modifications are as follows.

On page 14, we made revisions to the text:

“*Mucus clearing.* Oral drug delivery of large molecules is constrained by the degradation environment and malabsorption in the GI tract, e.g., peptides such as insulin and vancomycin have an oral bioavailability of less than 1%⁵⁸. To solve this problem, Traverso et al.²⁰ incorporated threaded features onto the outer surface of the capsule and introduced a motor-drive rotation to locally clear the mucus layer. This design innovation offers an effective mechanism for enhancement of drug absorption. Inspired by this, we implement similar functionality by developing a mucus-clearing MagCap with a threaded shell. It is noted that, the rotation of the capsule reported by Traverso et al.²⁰ is triggered in response to the pH change, necessitating no additional manipulation, thus constituting its most notable advantage, which shows great convenience and implementation in future applications. However, subject to its trigger mode, it is difficult to achieve the action at a specific site or a specific time. In contrast, magnetically actuated method needs to rely on an external magnetic field system, but can bring about a substantial enhancement of controllability.”

20. Srinivasan, S. S. *et al.* RoboCap: Robotic mucus-clearing capsule for enhanced drug delivery in the gastrointestinal tract. *Sci. Robot.* **7**, eabp9066 (2022).

58. Abramson, A. *et al.* An ingestible self-orienting system for oral delivery of macromolecules. *Science* **363**, 611–615 (2019).

Comment 3: There are numerous spelling and grammar issues, for example, in-vitro/in-vivo etc. are typically italicised, that need to be addressed throughout but my main concern with this work is the lack of detail in the main text which would impede the reproduction of the work. This includes the dimensions of the capsule, dimensions of the membrane, provenance of many of the materials used such as the NdFeB particles, the PDMS, the Ecoflex, PLA materials, the SQUID devices, The Fe₃O₄ particles, the magnetometer, UV-VIS spectrometer, lisinopril and so on.

Response 3: We sincerely apologize for the spelling and grammar issues you noticed, as well as the lack of detailed information that might hinder the reproducibility of our work. **We have thoroughly reviewed the concerns you raised and have made revisions** to ensure a comprehensive description of the materials used and parameters, enabling readers to better understand and reproduce our work.

The main modifications, apart from grammar, are as follows.

In the section of Methods, we added:

“Preparation of Magnetic Soft Valve

The main preparation process for the magnetic leaf and magnetic frame involves stirring, injection molding and heat demolding, laser cutting, and magnetization, as detailed in Supplementary Fig. S9.

The magnetic leaf is composed of silicone elastomer (Ecoflex 00–10) and NdFeB microparticles with an average size of 5 μm (MQP-15-7, Tianjin Magnequench Co., Ltd., China). In the experiments, these materials were sufficiently mixed in a container at a 1:1 mass ratio. Subsequently, the blend was mixed in a planetary mixer at 2,000 r.p.m. for 60 s, followed by defoaming at 2,200 r.p.m. for 45 s. Then, the homogeneous magnetic slurry was poured into a mold with a rectangular groove made of polytetrafluoroethylene and cured at room temperature for four hours. Finally, the prepared magnetic soft material was symmetrically folded and subjected to a pulsed magnetic field applied in the vertical direction for magnetization.

The magnetic frame is composed of polydimethylsiloxane (PDMS, SYLGARD 184, Dow Corning Co., USA) and NdFeB microparticles. In the experiments, the magnetic powder and PDMS were mixed in different mass ratios (7:3, 1:1, 3:7). Subsequently, the curing agent was added, and the mixture was thoroughly defoamed. The thoroughly blended magnetic mixture was then poured into the prepared mold and heated in an oven at 100°C for 1 hour. Finally, the prepared soft material for magnetic frame was subjected to a pulsed magnetic field applied in the vertical direction for magnetization.

It is noted that the rationale behind using different elastic materials for these two components can be explained as follows: Ecoflex 00-10, with its lower Young's modulus, allows for greater deformability, facilitating larger deformations of magnetic leaf. Conversely, to enhance the magnetic attractive force between the magnetic leaf and frame, a higher magnetic powder content is desirable for the magnetic frame. To address this requirement, the excellent flowability of PDMS proves advantageous in injection molding, enabling the attainment of a smooth and flat surface on the magnetic frame.”

On page 6, we made revisions to the text:

“After calibrating the magnetic field amplitude with a magnetometer (G93, Shenzhen Coliy Technology Development Co., Ltd., China), the critical magnetic fields for opening the valve are measured in these cases, as shown in Fig. 2c.”

On page 7, we made revisions to the text:

“First, we configure a push rod made by polylactic acid (PLA+, Shenzhen Esun Industrial Co., Ltd., China) with a known bottom area ($D=1.5$ mm) at the top surface of the precision weigher (AX324ZH, Changzhou OHAUS Instruments Co., Ltd., China) to exert force, propelling the magnetic leaf until it reaches a critical state of detachment from the magnetic frame.”

“Then in response to potential collision scenarios, the MagCaps containing the Ponceau S (Aladdin Biochemical Technology Co., Ltd., China) solution are placed in a beaker filled with water. The beaker is fixed on a shaker (OS-20, JOANLAB Equipment Co., Ltd., China) to simulate the vigorous peristalsis of the human GI under extreme conditions (Fig. 2f).”

On page 9, we made revisions to the text:

“We use methyl blue dye (Aladdin Biochemical Technology Co., Ltd., China) as the labeled drug and calculate the ratio between the sample group and the control group after drug release to estimate the actual drug release ratio.”

On page 11, we made revisions to the text:

“First, the gradient force and magnetic torque generated by a N35-grade permanent magnet with diameter of 50 mm and height of 30 mm (D50 mm×H30 mm, Beijing Jiuci Technology Co., Ltd., China) are used to control the multimodal motion of the capsule in the anatomical model of the human stomach, such as rotating (Stage I), sliding (Stage II), and rolling (Stage III).”

“This equipment mainly consists of a robotic arm (AUBO-i10, AUBO Robotics Technology Co., Ltd., China, Payload: 10 kg), a permanent magnet (Chengdu Juyu Magnetic Material Co., Ltd., China, D80 mm×H80 mm, N35), a handle controller, and a control panel, as displayed in Fig. 5a.”

On page 12, we made revisions to the text:

“After the rabbit ingests the MagCap, we use the gastroscope (2.9 mm CMOS electronic endoscope; Karl Storz, Inc., Germany) to enter its stomach along the esophagus to observe the motion trajectory of the capsule. The directional movement of the MagCap is carried out in the rabbit’s stomach using a permanent magnet (Chengdu Juyu Magnetic Material Co., Ltd., China, D80 mm×H80 mm, N35), where

it can achieve flexible directional rolling under the traction of a rotating magnetic field within 0-4 s, as shown in Fig. 6c.”

On page 14, we made revisions to the text:

“Subsequently, the absorbance of the collected intestinal fluid is assessed using a UV-vis spectrophotometer (UV-2600i, Shimadzu Co., Ltd., China), as shown in Fig. 7d-ii.”

On page 15 and 16, we made revisions to the text:

“The magneto-thermal part of the capsule consists of a mixture of Fe₃O₄ nanoparticles (50-300 nm, Aladdin Biochemical Technology Co., Ltd., China) and silica gel elastomer Ecoflex 00-10 (Beijing Angelcrete Art Landscaping Co., Ltd., China), and the heating processes for three mass fractions of 10%, 30% and 50% under an UHF magnetic field (38.5 kHz) generated by the induction heating machine (45 kW, Guangdong Taiguan Power Technology Co., Ltd., China) are shown in Fig. 8c-ii. It is shown that the higher the content of Fe₃O₄ particles, the higher the rate of heating, e.g., when the mass fraction is 50%, the highest temperature reached after heating for 60 s is equal to 98 °C by an infrared temperature imager (Ti400+, Fluke Shanghai Co., Ltd., China).”

On page 18, we made revisions to the text:

“The magnetic leaf is composed of silicone elastomer (Ecoflex 00–10) and NdFeB microparticles with an average size of 5 μm (MQP-15-7, Tianjin Magnequench Co., Ltd., China). In the experiments, these materials were sufficiently mixed in a container at a 1:1 mass ratio.”

“The magnetic frame is composed of polydimethylsiloxane (PDMS, SYLGARD 184, Dow Corning Co., USA) and NdFeB microparticles.”

On page 19, we made revisions to the text:

“For different experimental scenarios, two types of capsule shells were manufactured using UV-curable 3D printing technology, including transparent shells (Shenzhen

Chenno 3D Technology Co., Ltd., China) and MED610 shells (Guangzhou Chengxing Digital Technology Co., Ltd., China).”

“The magnetization of samples (magnetic composite materials) was measured with a superconducting quantum interference device (MPMS3-SQUID, Quantum Design Co., Ltd., USA) using the vibrating sample magnetometer option.”

On page 21, we made revisions to the text:

“The copper coil for generating the alternating magnetic field was fabricated by Yichang Zhuokai Technology Co., Ltd., China. Detailed information about the coil can be found in Supplementary Fig. S5e.”

“In addition, an AC power supply (SP300VAC5000W) manufactured by Beijing Quantian Technology Co., Ltd. was adopted to supply sinusoidal current to the coil with an iron core.”

Comment 4: Additionally, the results are often described in vague terms which impairs the readers' ability to judge the significance of the results, for example "This characteristic makes it easier to open the valve over long distances, as generating a sufficiently strong magnetic gradient is often challenging" - define long, define strong. Define what the low, medium and high frequencies each time for clarity. "This result suggests that the magnetic field generated by the coil has a high spatial resolution and can be utilized to achieve drug release or sampling in a specified area, e.g., different regions of the intestine, with a centimeter accuracy." - what is considered high spatial resolution in this context, how many centimeters, is it +/- 1cm, or +/- 10cm. "A certain amount of anesthetic is injected into the rabbit prior to the start of the experiments to alleviate its pain and ensure the safety of the experimenters," - how much is a certain amount? There are numerous similar examples such as this dotted throughout the text that need to be addressed.

Response: Thank you for your valuable comments. **We have revised the manuscript in accordance with your guidance, providing clearer descriptions to ensure a more comprehensive and precise presentation.** It is worth noting that in cases where specific values are not convenient to provide (such as magnetic field gradient),

we have re-expressed the relevant content in the revised manuscript to avoid ambiguity. The main modifications are as follows.

On Page 4, we made revisions to the text:

“Therefore, most magnetically driven capsules developed typically suffer from deficiencies, such as large size (>size #00 capsules, 8.53×23.3 mm)⁴¹, low ratio of the volume of loaded drug to the total volume of the capsule (low RDC)⁴², strict magnetic field requirements (settled direction or amplitudes exceeding 50 mT), and limited functionalities. Supplementary Table S1 contains a detailed list of capsule parameters. It's worth noting that, through innovations in capsule structure and materials, some extremely small capsules or those driven by low magnetic fields have been developed⁴³⁻⁴⁵. However, these advancements often come at the cost of sacrificing other key characteristics. For instance, the reported small capsules require a high magnetic field (> 100 mT) and have a low RDC (<0.1)⁴³.”

On Page 5 and 6, we made revisions to the text:

“It's worth noting that, unlike existing magnetic valves that rely on gradient magnetic forces for their opening mechanisms^{52, 53}, the proposed soft valve opens independently of the field gradient of the externally applied magnetic field. This independence is undoubtedly potent for remote control in practical applications, as the magnetic field gradient rapidly diminishes with increased distance between the capsule and the magnetic source. Additionally, the developed soft valve features automatic closure due to its inherent magnetic attractive force, whereas previous valves still required an external gradient magnetic field.”

On Page 8, we made revisions to the text:

“On the one hand, the specific magnetization structure of the magnetic soft valve allows the capsule to generally exhibit a single magnetic orientation characteristic (Fig. 1a-iii). Thus, the capsule is capable of global movement from Pos_1 to Pos_2 by applying a low-frequency magnetic field \mathbf{B}_1 (1~5 Hz). This global locomotion behavior is highly related to the magnetic torque τ_1 generated by the low-frequency

magnetic field \mathbf{B}_l and the residual magnetization \mathbf{M}_1 . On the other hand, the local deformation is controlled by applying a relatively high-frequency magnetic field (20~60 Hz), which can be attributed to magnetic torque τ_2 produced by the high-frequency magnetic field \mathbf{B}_h and the residual magnetization \mathbf{M}_2 .”

On Page 12, we made revisions to the text:

“This result suggests that the magnetic field generated by the coil has a relatively high spatial resolution (± 1 cm) and can be utilized to achieve drug release or sampling in a specified area, such as different regions of the intestine, with centimeter accuracy.”

“Prior to the start of the experiments, a certain amount of anesthetic (intraperitoneal injection of 3% sodium pentobarbital 3 ml/kg) is injected into the rabbit to alleviate its pain and ensure the safety of the experimenters, as shown in Fig. 6b.”

On Page 14, we made revisions to the text:

“In our application, the rotation of the MagCap accelerates the diffusion of the drug in the region by applying a medium-frequency magnetic field (5-10 Hz) after the completion of drug release (Supplementary Movie S11), as shown in Figs. 7c and Supplementary Fig. S8.”

On Page 15, we made revisions to the text:

“By combining low-frequency (1 Hz) and ultra-high-frequency (UHF, 20~80 kHz) magnetic fields, magnetic soft robots are able to deform and move while providing electrical or thermal energy, thereby offering the possibility to achieve more functions^{63, 64}.”

Comment 5: From a methodology perspective, some clarification on why the authors switched from using *ex-vivo* porcine tissue to an *in-vivo* small animal model (rabbit) would be welcome. A rabbit is not the typical animal model used for GI devices, usually a porcine model is used due to superficial similarities with the human GI tract.

Response: Yes, the use of a porcine model is more suitable for GI devices. Our perspective on this matter contains two main facets. Firstly, certain relevant

investigations in recent times have employed the rabbit as an example in the experiments, yielding noteworthy outcomes^{51, 57, 58}. Additionally, the capacity to conduct experiments in the rabbit stomach further demonstrates the compactness and miniaturization of our capsules.

On the other hand, it should be noted that pigs are substantially larger than rabbits. Consequently, generating a spatial magnetic field of 30 mT in the pig stomach model presents more demanding requirements for both power supply and space. Regrettably, we were unable to pursue further experimentation due to the current experimental circumstances. We acknowledge that the experimental design holds certain inadequacies, and we intend to enhance the coil design in the future. Nevertheless, generating 30mT in a medical environment is both practical and secure based on previous research.

To emphasize our considerations in animal selection during the revision, we included the following information.

On Page 12, we added:

“To further test the motility of our developed MagCaps in a confined small space environment and verify their feasibility for biomedical applications, we conducted an *in-vivo* drug delivery test in a rabbit's stomach, where similar rabbit models were previously used in drug delivery and release studies within the GI tract^{51, 57, 58}”

51. Yang, X. *et al.* An agglutinate magnetic spray transforms inanimate objects into millirobots for biomedical applications. *Sci Robot* **5**, eabc8191 (2020).

57. Cai, L. *et al.* Rocket-Inspired Effervescent Motors for Oral Macromolecule Delivery. *Advanced Materials* **35**, 2210679 (2023).

58. Wen, S. *et al.* Ca-Alginate-Based Janus Capsules with a Pumping Effect for Intestinal-Targeted Controlled Release. *Engineering* **24**, 114–125 (2023).

Response to the reviewer #3:

General Comment: The authors were able to generate a capsule for drug delivery, sampling, and locomotion using a novel magnetic leaf valve design. For additional functionality, onboard lights, heat generating materials, and mucus clearing shell were incorporated to the compact device. By adjusting the frequency of an external rotating magnetic field either above or below the capsule's step-out frequency, sampling/drug delivery and locomotion were independently addressed.

Response: We greatly appreciate your general comment and will provide point-by-point response to the specific comments in the following part.

Comment 1: While the capsule's multifunctionality was clearly demonstrated, more discussion on valve modeling, locomotion modeling, and limitations to their capsule and locomotion strategy could have been provided.

Response 1: Apologies for the lack of clarity in our previous description. In response to the reviewer's suggestions, we have included further discussion in the revised manuscript. Specifically, **additional content has been added including Supplementary Text S3 (Fluid-solid coupling analysis) and Text S4 (Locomotion mechanisms)** pertaining to valve and locomotion modeling, respectively. Additionally, within the Discussion section, **a more detailed analysis has been provided concerning the limitations** associated with the capsule, locomotion strategy and other aspects. The main modifications are as follows.

In Supplementary Materials, we added:

Text S3. Fluid-solid coupling analysis

In Fig. 3d of the manuscript, the experiment was conducted by placing the MagCaps within a receptacle filled with liquid, and subsequently subjecting them to a high-frequency (10-60 Hz) magnetic field. Throughout the experiment, the magnetic leaf within the MagCaps underwent periodic bending, showcasing a characteristic fluid-solid coupling phenomenon. We contend that simulating this phenomenon will aid in comprehending the functional principles of the MagCaps.

In this study, finite element software COMSOL was utilized for analysis, and a 2D model was developed. Despite the distance between the 2D model simulation and the actual experiment, it remains valuable for qualitatively analyzing changes in the flow field. In the simulation, the magnetic valve is divided into i equivalent elements, and the distributed magnet torque is substituted by the cumulative magnetic torque, achieved through two tangential surface forces (in $\text{N}\cdot\text{m}^{-2}$) relative to the boundary frame^{S6}.

$$F_i = M_x^i B_y - M_y^i B_x \quad (\text{S4})$$

where M_x^i and M_y^i represent the average magnetization of the element in the x_g - and y_g -axes, and B_x and B_y are the components of the magnetic flux density magnitude B in the x_g - and y_g -directions, respectively.

The flow field surrounding the magnetic leaf is unsteady and nonuniform. Therefore, we solve the complete Navier-Stokes equation involving the inertial, convection, pressure, and diffusion terms. The fluid is assumed to be Newtonian and incompressible. Considering the mass and linear momentum balance, the physical behavior of the fluid is captured through the following equations:

$$\nabla \cdot \mathbf{u} = 0 \quad (\text{S5})$$

$$\rho_f [\dot{\mathbf{u}} + (\mathbf{u} \cdot \nabla) \mathbf{u}] = -\nabla p + 2\mu \nabla \cdot \mathbf{D} \quad (\text{S6})$$

Here, p represents the scalar pressure field, \mathbf{D} denotes the rate of deformation tensor, and \mathbf{u} is the velocity field. μ and ρ_f represent the fluid viscosity and density, respectively.

Utilizing the aforementioned model, we can simulate the fluid effects arising from the periodic deflection of the magnetic leaf induced by an alternating magnetic field (**Fig. S13**). From the onset to the quarter cycle (0-0.25 T), the magnetic leaf deflects upwards, leading to an increase in pressure within the capsule cavity. The liquid inside the capsule then extends in a counterclockwise flow, with a portion of the internal liquid flowing out through the channel. As the deflection angle of the magnetic leaf

increases (0.5 T), liquid outside flows into the capsule along the channel's side. Simultaneously, a vortex ring forms beneath the magnetic leaf, initiating a rapid mixing process between the internal chamber liquid and the inflowing external liquid. In the latter half of the cycle (0.5-1 T), the magnetic leaf gradually approaches closure. During this phase, a portion of the mixed liquid enters the interior of the capsule, while another portion exits the capsule. For instance, considering a frequency of 30 Hz, the MagCap undergoes 30 analogous cycles within a second under the influence of a high-frequency magnetic field. This phenomenon establishes the foundation for the release and sampling within the MagCaps.

Fig. S13. Fluid exchanging processes within a capsule under the action of an applied magnetic field.

Text S4. Locomotion mechanisms

The movement of the MagCaps along the S-bend track (Fig. 4b) can be considered as planar motion, and it can be roughly categorized into two modes (**Fig. S14**): rolling and turning. The axial magnetization of the magnetic frame and the radial symmetric magnetization of the magnetic leaf provide the MagCap with an overall single magnetic orientation (\mathbf{M}_1), which lays the foundation for the MagCap's controllable mobility. In the presence of a uniform magnetic field, the MagCap will roll until the magnetization direction of the magnetic frame inside the capsule aligns with the direction of the magnetic field. In cases where the applied uniform magnetic field rotates, the capsule will simultaneously rotate and roll either forward or backward.

When the capsule requires movement from Pos_1 to Pos_2 (**Fig. S14-b**), with \mathbf{B}_{x1} representing the initial desired magnetic field and \mathbf{n} as the direction vector of the rotation axis, the required magnetic field \mathbf{B}_{y1} to induce the roll can be expressed as follows^{S7}:

$$\mathbf{B}_{y1} = \mathbf{R}(\mathbf{n}, \varphi_{12}) \mathbf{B}_{x1} \quad (\text{S7})$$

where $\mathbf{R}(\mathbf{n}, \varphi_{12})$ is a rotation matrix that rotates angle φ_{12} about axis \mathbf{n} . When \mathbf{n} is in X-Y plane and orthogonal to the moving direction of the capsule, the capsule can keep moving forward by continually updating \mathbf{B}_{y1} .

Hence, Based on the theoretical analysis described above and Eq. (S7), in the rolling mode depicted in **Fig. S14-b**, the magnetic torque τ_{12} acting on the MagCap is expressed in terms of the driving magnetic field \mathbf{B}_{y1} and the residual magnetization strength M_1 , respectively, as

$$\tau_{12} = M_1 \times \mathbf{B}_{y1} \quad (\text{S8})$$

where \mathbf{B}_{y1} is the high frequency alternating magnetic field generated by an external coil, and m_1 is the residual magnetization strength retained by the magnetized NdFeB particles in the magnetic frame. For this reason, the capsule can be tumbled around the Z-axis by generating a rotating magnetic field in the XOY plane. The specific drive waveform is illustrated in **Fig. S15a**, with the signal being output from the STM32 (**Fig. S6d**) and subsequently normalized.

Furthermore, we can introduce a magnetic field along the Z-axis into the 2D magnetic field, as depicted in **Fig. S14c**. Therefore, the magnetic torque τ_{34} responsible for turning the capsule to the left can be calculated using the following equation.

$$\tau_{34} = M_1 \times \mathbf{B}_{z4} \quad (\text{S9})$$

The magnetic field waveform that actuates the capsule is depicted in **Fig. S15b**. The rotation of the MagCap can be controlled by changing the direction of the rotating magnetic field. By detecting the push direction from the handle pusher, the MagCap can roll towards the 45-degree direction using the waveform in **Fig. S15c**.

Fig. S14. Multimodal locomotion mechanisms of capsules. (a) Illustration of two typical locomotion modes of Magcaps. (b) Left view of the rolling mode. (c) Top view of the turning mode.

Fig. S15. Driving magnetic field waveforms of the MagCap in various cases. (a) Waveform of the driving magnetic field for rolling mode. (b) Waveform of the driving magnetic field for the turning mode. (c) Waveform of the driving magnetic field for MagCap rolling towards the 45-degree direction.

In the Discussion section, we made revisions to the text:

“We clearly validated multiple potential application functions of the developed capsules; however, future work should combine specific pathology experiments to optimize the capsule structure design and to evaluate the advantages of these MagCaps. For instance, compared with traditional methods, quantitative analysis with respect to reducing the drug dosage, the medication frequency, and the side effects due to the use of locally targeted drug release should be discussed; It is necessary to determine the optimal size of capsules based on the required drug release or sampling volume in practical application scenarios; It is also interesting to leverage the sampling function of MagCaps to investigate the correlation between intestinal flora and diseases across different intestinal regions. Additionally, regarding further improvements in capsule technology or performance, the following aspects are worth considering. Firstly, concerning the overall motion of MagCaps, as observed from Figure 5, it is evident that the combined application of magnetic torque and gradient magnetic forces enriches their overall motion patterns. However, predicting and precisely controlling the capsule's motion under the simultaneous influence of these two driving forces still requires further in-depth research. Secondly, the integration of additional actuation sources, such as built-in motor²⁰ or mechanical springs⁶⁰, into capsule design could complement magnetic actuation strategies, aiming to simplify the external magnetic sources and actuation strategies. Thirdly, the emergence of capsule endoscopy has enabled comprehensive visualization of the entire GI tract, showing promising potential as an effective clinical tool for monitoring gut health^{6, 65}. Therefore, the integration of the imaging module from existing capsule endoscopy into the MagCaps could significantly enhance real-time monitoring capabilities during capsule operation. Lastly, it is favorable to know the position and orientation of the capsule because it can guide the application of magnetic field for the desired motion, a key factor for this was the introduction of real-time localization techniques⁶⁶. Considering the medical CT imaging methods usually exist radiation for patients and health-care workers, we can account for some assisted positioning technologies for the long-term dynamic positioning of MagCaps in the GI environment, such as ultrasound⁶⁷ and magnetic^{68, 69} assisted localization technology.”

Comment 2: Moreover, different magnetic actuation systems were used for different tests (i.e., Helmholtz coil systems, permanent magnets, and coil system on robot arm etc.). It was unclear at times which actuation system was being used and how capsule and valve behaviour measured with the 3D coil system at the beginning of the paper would change with the use of permanent magnets and 2D coil at the end of the paper.

Response 2: Apologies for the lack of clarity in our previous description. **In the revised manuscript, we have provided explicit clarification on the specific magnetic actuation systems employed in different experimental scenarios**, where a new table (Table S3) has been added in the Supplementary Material.

In addressing the second question, we provide the following explanation. Initially, in this study, we conducted experiments employing the Helmholtz coil to quantitatively examine the dynamics of MagCaps. This initial investigation allowed us to analyze the influence of independent variables, such as amplitude and frequency, on the behavior of the MagCaps. Subsequently, recognizing the inconvenience of using Helmholtz coils within animal experiments (*In vitro* and *In vivo*), we employed a robotic arm equipped with a permanent magnet for generating low-frequency magnetic fields and a coil for generating high-frequency magnetic fields in practical scenarios. A key distinction between these magnetic sources and the Helmholtz coil lies in their gradient effect. It is worth noting that, our emphasis in this article predominantly focuses on the influence of magnetic torque, including capsule rolling and valve opening, which occurs in both uniform and gradient magnetic fields. However, the presence of a gradient adds an additional force to the overall motion of the capsule, affecting its practical movement. For instance, as shown in Fig. 5b, our observations suggest that the gradient force can induce a sliding motion pattern in the capsule, while also enhancing its positioning function. In fact, the introduction of a gradient makes the capsule's motion more richer. However, the corresponding physical models and control also becomes more complex. Therefore, for situations with high requirements for control accuracy, this is an issue that requires further in-depth investigation, as indicated in the revised manuscript (see **Discussion**).

The main modifications are as follows.

Table S3. Magnetic actuation systems used in different experimental scenarios.

Figure Number	Type of Magnetic Actuation Systems	Magnetic Type	Magnetic Field Amplitude	Magnetic Field Frequency
Fig. 2b-d	Biaxial Helmholtz coil	Stable	0-40 mT	-
Fig. 3b	Triaxial Helmholtz coil	Rotating (X-Y)	10 mT	0.5-20 Hz
Fig. 3c, d	Self-made coil	Sinusoidal	30 mT	10-30 Hz
Fig. 4b, c	1. Triaxial Helmholtz coil 2. self-made coil	Rotating (X-Y-Z)	1. 10 mT 2. 30 mT	1. 1Hz 2. 30 Hz
Fig. 4d	1. Permanent magnet (D50×L20 mm, N35) 2. Self-made coil	1. Rotating 2. Sinusoidal	1. - 2. 30 mT	1. <1 Hz 2. 30 Hz
Fig. 5a-c	1. A permanent magnet (D80×L80 mm, N35) on a 6-DOF arm 2. Self-made coil	1. Rotating 2. Sinusoidal	1. - 2. 30 mT	1. <1 Hz 2. 30 Hz
Fig. 5e-i	A poly-magnetic coil on a 6-DOF arm	Sinusoidal	30 mT	30 Hz
Fig. 6	1. Permanent magnet (D80×L80 mm, N35) 2. Self-made coil	1. Rotating 2. Sinusoidal	1. - 2. 30 mT	1. <1 Hz 2. 30 Hz
Fig. 7a, b	Self-made coil	Sinusoidal	15 and 25 mT	30 Hz
Fig. 7c, d	Self-made coil	Sinusoidal	15 mT and 30 mT	5-10 Hz and 20-40 Hz
Fig. 8a, b	1. Permanent magnet (D50×L20 mm, N35) 2. Wireless Powered Transmitter Coil	1. Rotating 2. Sinusoidal	1. - 2. -	1. <1 Hz 2. 35.7 kHz
Fig. 8c, d	1. Permanent magnet (D50×L20 mm, N35) 2. Self-made coil 3. Wireless heating coils	1. Rotating 2. Sinusoidal 3. Sinusoidal	1. - 2. 30 mT 3. -	1. <1 Hz 2. 30 Hz 2. 35.7 kHz

Comment 3: The authors identified capsule size as one of the limitations in the introduction (line 78 – 83). Capsules with smaller sizes than capsule 00 (D 8.53 mm x L23.3 mm) have been previously reported with potential for both drug delivery, sampling, and locomotion.

(1) P. Shokrollahi et al., Blindly Controlled Magnetically Actuated Capsule for Noninvasive Sampling of the Gastrointestinal Microbiome, IEEE/ASME Transactions

on Mechatronics, 26, 5, 2616-2628, 2021.

(2) A. Abramson, et al. An ingestible self-orienting system for oral delivery of macromolecules. Science, 363, 611–303, 2019.

(3) Jiachen Zhang et al.,Voxelated three-dimensional miniature magnetic soft machines via multimaterial heterogeneous assembly.Sci. Robot.6,eabf0112,2021.

Moreover, size directly correlates with the volume of drug delivered or sample retrieved. What is the average drug volume or sample volume required for your applications?

Response 3: Thank you for your reminder. Indeed, we previously mentioned capsule size as a limitation in the manuscript, while it mainly reflects the majority of cases. **We have revised the corresponding description to clarify this point more accurately.** Additionally, in Table S1, we have highlighted large-size capsule in red.

Regarding the second question, our current work primarily focuses on analyzing the multifunctional aspects of the capsules rather than assessing the specific drug volume or sampling volume. This is because such evaluations require case-specific considerations, which we acknowledge as a part of future work, as explicitly mentioned in the section of **Discussion**. We appreciate your valuable input on this matter, prompting us to consider a metric to evaluate the relative drug-carrying capacity of the capsules, namely the ratio of the volume of loaded drug to the total volume of the capsule (RDC). Consequently, in the revised manuscript, we have introduced this parameter as a characteristic of the capsules and presented it in **Supplemental Table S1**. Notably, our capsules demonstrate a remarkably high RDC level.

The main modifications are as follows.

On page 4, we made revisions to the text:

“Therefore, the most of developed magnetically driven capsules typically suffer from deficiencies such as large size (>size #00 capsules, 8.53×23.3 mm)⁴¹, low ratio of the volume of loaded drug to the total volume of the capsule (low RDC)⁴², strict magnetic

field requirements (settled direction or high amplitude more than 50 mT) and simple functions. A detailed list of capsule parameters is available in Supplementary Table S1. It's worth noting that, through innovations in capsule structure and materials, some extremely small capsules or those driven by low magnetic fields have been developed⁴³⁻⁴⁵. However, they typically come at the cost of sacrificing other key characteristics. For instance, the reported small capsules require a high magnetic field (> 100 mT) and have a low RDC (<0.1)⁴³. The capsules driven by low magnetic fields also face issues such as complex structures or single functionality without active control^{44, 45}. Additionally, significant changes in the external structures of these capsules occur during drug release or sampling process, which could allow considerable resistance to the functionality of the capsules in an unstructured and narrow environment (such as the small intestine).”

Revised Table S1. Comparison of the proposed capsules with the ones previously reported in the literature.

References	Actuated source	Built-in magnetic source	Targeted transport	Delivery/sampling	Magnetic field or force strength for driving	Size (mm)	Delivery/Sampling Volume (mm ³)	RDC	Overall or local deformation	Expandable functions
[7]	Motor	—	×	√/×	Motor power (~250 mW)	D8.2*L24.5	342.6	0.13	Local	■ Mucus-clearing
[8]	MF & motor	Permanent magnet	√	×/√	—	D11.6*L32	350	0.1	Local	■ Camera
[9]	Spring	—	×	√/×	Mechanical force 9 N	D9*L15	—	—	Local	■ Oral delivery for macromolecules
[10]	Battery	—	×	×/×	—	D12*L27	—	—	—	■ Photodynamic therapy for Helicobacter pylori infection
[11]	MF ¹	—	×	×/×	—	D15	—	—	—	■ Photodynamic therapy for cancer therapy
[12]	MF	Permanent magnet	√	×/√	0.55 N	D15*L32	0.35	6.2×10 ⁻⁵	Overall	■ Biopsy
[13]	MF & Ultrasound	Magnetic core	√	√/×	1.5-7 T	D5	—	—	Local	■ MRI image
[14]	MF	Permanent magnet	√	√/×	150 mT	D16*L32.5	800	0.12	Overall	—
[15]	MF	Permanent magnet	×	√/×	70 mT	D20*L40	800	0.06	Overall	—
[16]	MF	Permanent magnet	√	×/√	44.4 mT	D13*L32	5	0.001	Local	■ Camera ■ Biopsy
[17]	MF	Permanent magnet	√	×/√	Magnetic force 0.6 N	D17.8*L24.6	—	—	Overall	■ Fine-Needle Biopsy
[18]	MF	Soft magnet	√	√/×	375 mT	D12*L33	780	0.24	Overall	—
[19]	MF	Permanent magnet	√	√/×	—	D14*L28.9	2045	0.46	Local	■ Multi-drug delivery
[20]	MF	Permanent magnet	×	×/√	6 mT	D12.4*L26	1500	0.48	Overall	—
[21]	MF	Permanent magnet	×	×/√	15 mT	D8*L11	42	0.07	Overall	—
[22]	MF	Permanent magnet	-	×/√	103.5 mT	D9*L24	1	0.005	Local	■ Biopsy
[23]	MF	Magnetic composite (Cobalt & NdFeB)	-	-	295 mT	D0.5*L0.82	6.44×10 ^{-3#}	0.039 [#]	Overall	—
	MF	Magnetic composite	√	√/√	100 mT	D0.5*L0.45	2.34×10 ^{-3#}	0.026 [#]	Overall	—
Our work	MF	Double-layer magnetic composites	√	√/√	30 mT	D5.8*L13 D8.4*L19.5	121.6 457.6	0.35 0.42	Local	■ Multi-drug delivery ■ Mucus-clearing ■ Light-assisted therapy ■ Thermal-assisted therapy

#: It is not given directly in the original article but is obtained by our estimation.

MF: Magnetic Field; 1: The magnetic field is only used as a wireless power supply.

RDC: Ratio of the volume of loaded drug or sampling to the total volume of the capsule.

Overall deformation: When the capsule fulfills the function of drug delivery or sampling, the overall structure of the capsule will change, such as separating into two parts or undergoing compression deformation.

Local deformation: When the capsule fulfills the function of drug delivery or sampling, the overall external structure and shape of the capsule remain unchanged, only the internal structure changes.

[23]: This type is technically capsule-shaped rather than a true capsule.

Red font: Indicating that the capsule size proposed in the aforementioned literature is larger than Size 00 capsules (8.53×23.3mm) or the required driving magnetic field is higher than 50mT.

Comment 4: For magnetically actuated capsules, capsule size would also affect the magnetic torque available for drug delivery and locomotion. What would be the smallest capsule size possible that would still ensure successful and sufficient drug delivery with your current magnetic actuation systems?

Response 4: We fully acknowledge the reviewer's insightful comment regarding the size of MagCaps. In our study, the smallest capsule size we have explored is D5.8 mm *L13 mm. It is indeed feasible to create smaller MagCaps, as magnetic soft composites have demonstrated potential for miniaturization in various other domains. The reason we haven't extensively covered smaller sizes in our work has two aspects.

Primarily, further miniaturization encounters challenges associated with the precision of printing biocompatible materials. Secondly, in the context of gastrointestinal capsules, the presently available sizes are entirely suitable, even for young children (the small intestine has a diameter of 7mm at its narrowest point). Additionally, as mentioned above, it's crucial to note that capsule size should be determined or optimized based on specific needs, considering the required sampling or release volumes. **Similar to the previous question, this aspect has been discussed in the manuscript's Discussion section.**

Comment 5: What would be classified as high magnetic fields or gradients? Would the authors be able to provide a quantitative number?

Response 5: Sorry for our vague expression. In the revised manuscript, we have clarified that "high magnetic fields" refer to magnetic field strengths exceeding 50 mT. Acknowledging the pivotal role of magnetic torque as the primary driving force for the capsule and considering that gradient information might not be universally pertinent across various magnetic capsule applications, we have emphasized the magnetic field strength rather than the gradient and revised the relevant descriptions.

On page 4, we made revisions to the text:

“Therefore, the most of developed magnetically driven capsules typically suffer from deficiencies such as large size (>size #00 capsules, 8.53×23.3 mm)⁴¹, low ratio of the volume of loaded drug to the total volume of the capsule (low RDC)⁴², strict magnetic field requirements (settled direction or high amplitude more than 50 mT) and simple functions.”

Comment 6: For magnetic characterization using the SQUID, what were the samples you were measuring? Was it the NdFeB particles or composite material?

Response 6: Apologizing for the confusion caused by our vague expression. In our magnetic characterization using the SQUID, the samples measured were magnetic composite materials rather than NdFeB particles. This distinction has been clarified in the revised manuscript (see “**Magnetic characterization**”).

Comment 7: For the capsule sealing tests, why was 100 rpm chosen? Does it correlate with the abdominal pressures or forces in the digestive tract?

Response 7: Thank you for providing constructive feedback regarding the choice of 100 rpm for the capsule sealing tests. We aimed to simulate a vigorous oscillation intensity significantly surpassing the frequency of gastrointestinal contractions in normal humans (9-12 contractions per minute). **This rationale has been explicitly clarified in the revised manuscript.**

Additionally, **prompted by your suggestion, we have included a specific test to evaluate the pressure tolerance of the capsule.** As a result, we have introduced a corresponding data figure (Fig. 2e) in the revised manuscript. The outcomes demonstrate that MagCaps with a magnetic frame mass fraction exceeding 50% can effectively maintain a seal under gastrointestinal pressure.

The main modifications are as follows.

On page 7, we made revisions to the text:

“The shaker speed is set to 100 rpm (Supplementary Fig. S5 and Movie S1), significantly exceeding the frequency of gastrointestinal contractions in normal humans (9-12 contractions per minute⁵⁵).”

“First, we configure a push rod made of polylactic acid (PLA+, Shenzhen Esun Industrial Co., Ltd., China) with a known bottom area ($D=1.5$ mm) at the top surface of the precision weigher (AX324ZH, Changzhou OHAUS Instruments Co., Ltd., China) to exert force, propelling the magnetic leaf until it reaches a critical state of

detachment from the magnetic frame. The force value at this moment is measured and converted into pressure, as shown in Fig. 2e. Some studies⁵⁴ have concluded that the pressure of the digestive phase does not exceed 25 mm Hg (3.33 kPa). Therefore, MagCaps with the magnetic frame mass fraction exceeding 50% can maintain a seal under gastrointestinal pressure.”

Revised Fig. 2e Critical pressures required for the magnetic leaf detachment from the magnetic frame with varying magnetic powder mass fractions.

Comment 8: In the *in-vivo* studies, a permanent magnet was used to control the capsule which would have both magnetic gradient and field. How would you account for field gradients that may affect the precise control of the capsule?

Response 8: We acknowledge and share your concern regarding the potential impact of field gradients on the precise control of the capsule. As mentioned in our response to the previous question (**comment-2**), employing permanent magnets offers convenience and introduces gradient-related opportunities for diverse capsule movements. However, we concur that developing control strategies for precise manipulation becomes challenging due to the increased complexity in the associated physical models and motion mechanisms. Our ongoing research confirms the

capsule's successful performance in local and overall movements under conditions involving magnetic field gradients. Nonetheless, achieving fine-tuned control and precise actuation in these magnetic fields warrants further in-depth exploration, which has been mentioned in the section of **Discussion** in the revised manuscript.

In Discussion section, we added:

“Firstly, concerning the overall motion of MagCaps, as observed from Figure 5, it is evident that the combined application of magnetic torque and gradient magnetic forces enriches their overall motion patterns. However, predicting and precisely controlling the capsule's motion under the simultaneous influence of these two driving forces still requires further in-depth research.”

Comment 9: What type of polymer material is used in MED610?

Response 9: MED610 is a transparent and biocompatible PolyJet photopolymer manufactured by Stratasys Ltd. **This information has been included in the revised manuscript, and a reference is added.** It is noted that MED610 is a specific product by Stratasys Ltd. designed for 3D printing purposes, and we believe such clarification is clear.

On page 18 and 19, we made revisions to the text:

“For different experimental scenarios, two types of capsule shells have been manufactured using UV-curable 3D printing technology. These include transparent shells (Shenzhen Chenno 3D Technology Co., Ltd., China) and MED610 shells (Guangzhou Chengxing Digital Technology Co., Ltd., China). The former is beneficial for observing the deformation of the magnetic leaf and its interaction with the magnetic frame in dynamic studies. The latter, a PolyJet photopolymer manufactured by Stratasys Ltd., is more biocompatible⁷⁰ and ensures capsule's resistance to corrosion by gastric acid and other substances in biological validation experiments. For detailed information, please refer to Supplementary Fig. S10, demonstrating that the capsule does not soften when immersed in acid.”

Comment 10: What was the size and grade of the permanent magnet in Fig. 4d? Would you be able to provide more details on the various coil systems and robot arm used?

Response 10: Sorry for our unclear description. In the revised manuscript, **we have provided the size and grade of the permanent magnet in Fig. 4d.** Additionally, **we have enriched the information regarding the coil systems and the robot arm** used in our study. Specifically, we have provided the purposes of the various magnetic actuation systems in **Table S3**, and the structure parameters of these coils in **Table S4**. The size and grade of the permanent magnet in Fig. 4d are D50 × L30 mm and N35 respectively.

The main modifications are as follows.

On page 11, we made revisions to the text:

“First, the gradient force and magnetic torque generated by a N35-grade permanent magnet with diameter of 50 mm and height of 30 mm (D50 mm×H30 mm, Beijing Jiuci Technology Co., Ltd., China) are used to control the multimodal motion of the capsule in the anatomical model of the human stomach, such as rotating (Stage I), sliding (Stage II), and rolling (Stage III).”

“This equipment mainly consists of a robotic arm (AUBO-i10, AUBO Robotics Technology Co., Ltd., China, Payload: 10 kg), a permanent magnet (Chengdu Juyu Magnetic Material Co., Ltd., China, D80 mm× H80 mm, N35), a handle controller, and a control panel, as displayed in Fig. 5a.”

In Supplemental Materials, we added:

Table S4. Structural features of various coils. (All units are in millimeters)

Figure	Various coils	Components	Structural parameters	Value
Fig. S5a	Biaxial Helmholtz coil	X-axis coils	Inner diameter	128
			Outer diameter	240
		Y-axis coils	Height	150
			Inner diameter	262

			Outer diameter	400
			Height	227
		X-axis coils	Inner diameter	390
			Outer diameter	500
			Height	285
		Y-axis coils	Inner diameter	260
Fig. S5b	Triaxial Helmholtz coil		Outer diameter	370
			Height	210
		Z-axis coils	Inner diameter	155
			Outer diameter	242
			Height	140
			Inner diameter	40
Fig. S5e	Self-made coil	Z-axis coils	Outer diameter	200
			Height	60

Comment 11: The dual-mode sampling, light and heating capsule experiments appears to be missing in the methods section.

Response 11: Regarding the term "dual-mode sampling", we believe the reviewer might be referring to "dual-drug release". As the experimental setup and methodology employed for dual-drug release remain consistent across both the "Sealing-ability test method" and the "Drug release rate test" sections, we have made an improvement in the revised manuscript by adding some supplements to the "Dual-drug release" section in the main text.

Regarding the experiments involving light and heating capsules, **prompted by your reminder, we have added a description of this aspect in the Methods section.**

On page 22, we added:

“Wireless light and heating experiment setup. In the wireless light experiment, we used a permanent magnet (50 mm × 30 mm, N35) and a homemade coil (Fig. S6e) to generate a low-frequency (1 Hz) and ultra-high-frequency (37.5 kHz) magnetic field, respectively. In the wireless heating experiment, the magneto-thermal part of the capsule was heated by an ultra-high-frequency magnetic field (38.5 kHz) generated by an induction heating machine (45 kW, produced by Guangdong Taiguan Power Technology Co., Ltd., China), as shown in Fig. 8c-ii. Meanwhile, the temperature of the area heated by the ultra-high-frequency magnetic field was measured by an infrared temperature imager (Ti400+, Fluke Shanghai Co.,Ltd., China).”

Comment 12: Fig. 1a (i) and Fig. 1a (ii) – It was not clear where the drug exits the capsule, perhaps the author can label the capsule opening.

Response 12: Sorry for our unclear description. **We have added labels to indicate the channels** through which the drug exits the capsule in Fig. 1a.

Revised Fig. 1a Structure of the MagCap and drug injection process.

Comment 13: Fig. 1a – It was unclear how the magnetic leaf generates a ‘U’ shape under an external magnetic torque. Authors should show that the middle of the leaf is fixed to the center to enable consistent generation of the ‘U’ shape.

Response 13: Thank you for your valuable feedback. Indeed, as you mentioned, the middle of the leaf is fixed at the center to ensure the consistent formation of the 'U' shape. **In response to your suggestion, we have added the labeling to clearly indicate the center of fixation in Fig. 1b.**

Revised Fig. 1b Schematic diagram showing the opening and closing process of the soft valve under an external magnetic field.

Comment 14: Fig. 1f – The figure label should describe what capsules 1 – 6 are and the scale bar should be labelled.

Response 14: Sorry for our unclear description. **In the revised manuscript, the capsules in Fig. 1f have been labelled and the scale bar have be added** (see Fig. 1f and the corresponding figure caption).

Revised Fig. 1f Photographs of the developed six MagCaps (Miniature-Cap1, Visualisation-Cap2, Dual-module-Cap3, Threaded-Cap4, Light-assisted-Cap5, Thermal-assisted-Cap6). The dimensions, functions and other detailed information of these MagCaps can be found in Supplementary text S1, Table S2, and Fig. S1). Scar bar: 10 mm.

Comment 15: Line 128 – 129, “It can be concluded that the NdFeB particle content is positively correlated with the magnetic gradient force...” The sentence should be rephrased as positive correlation sounds like a regression analysis was performed.

Response 15: Thank you for your suggestion. In the revised manuscript, **the sentence has been modified** as “It can be concluded that the NdFeB particle content has a positive relation with the magnetic gradient force...” (on Page 6).

Comment 16: What is the definition of the valve’s opening magnetic field? Is there a specific opening angle this corresponds to?

Response 16: In our work, the valve's opening magnetic field is defined as the magnitude of the magnetic field at which an externally excited magnetic field causes the magnetic leaf to detach from the magnetic frame. This has been clarified in the revised manuscript. It should be noted that, due to the presence of gradient magnetic forces between the magnetic leaf and magnetic frame, the magnetic leaf undergoes a

sudden change in its opening angle under the action of critical external magnetic field (see Fig. 2d), making it difficult to pinpoint a specific opening angle for judgment.

Fig. 2d. Measured bending angles with different magnetic powder contents under the static magnetic field.

On Page 6, we made revisions to the text:

“For instance, at a mass fraction of 30% NdFeB particles, the valve's opening magnetic field—indicating the magnitude of the externally applied magnetic field at which the magnetic leaf detaches from the magnetic frame—is approximately 6.8 mT.”

Comment 17: Fig. 2a – Magnetic moment usually has units of $A \cdot m^2$. Do you mean magnetization?

Response 17: Sorry for the inaccurate description. Yes, it should be magnetization.

This point has been modified in the revised manuscript.

Revised Fig. 2a. Magnetic hysteresis loops of the used magnetic soft composites containing NdFeB microparticles and PDMS, the mass fractions of NdFeB are 30%, 50% and 70%, respectively.

Comment 18: Fig.2e – Is α_2 the same angle as α_1 in Fig. 2d?

Response 18: Yes, as you mentioned, both α_1 (Fig. 2d) and α_2 (Fig. S4) represent the bending angle under a static magnetic field. **In the revised manuscript, to avoid ambiguity, they have been unified as α_1 .**

Comment 19: Fig. 2d and Fig. 2e seems a bit redundant as it is showing similar trends of bending angle vs. magnetic field.

Response 19: Thank you for your valuable suggestion. While there is some overlap in the information presented in the two charts, they differ in their primary focus. The former primarily aims to illustrate magnetic leaves bending at nearly the same angle under identical magnetic field conditions after detachment from the magnetic frame in the experiment. In contrast, the latter focuses on comparing simulation and experimental results of magnetic lobe deformation in the absence of magnetic locking. Nonetheless, **we appreciate and accept your valuable suggestion to relocate Figure 2e to the supplementary material and to include a new figure depicting the experiment involving pressure.**

Comment 20: Fig. 3d – What is α_3 ? Is it the same angle as α_1 and 2? What are the dots in Fig. 3d(ii)?

Response 20: Sorry for our unclear description. **We have clearly defined the angles that emerge in different places.** In the revised manuscript, α_1 means the bending angle under a static magnetic field (previous α_1 and 2), and α_2 means the maximum bending angle under a sinusoidal alternating magnetic field. Additionally, the dots in Fig. 3d(ii) represent the actual measurements obtained from multiple repetitions, **which has been pointed out in the revised Fig. 3d (ii).**

Comment 21: Is there a resonant frequency at which your valves operate at?

Response 21: Thus far, we have not observed a similar phenomenon in our experiments.

Comment 22: Movie S2 – were the capsule actuated by a continuous rotating magnetic field for the same duration? Why does the last frame in the movie look different from Fig. 3b (i.e., the capsule's distances at 5 Hz and 10 Hz in the video vs.

Fig. 3b)?

Response 22: Sorry for our mistake. Yes, as you mentioned, the capsule in Movie S2 was actuated by a continuous rotating magnetic field for the same duration. **This mistake has been modified in the revised Movie S2.** In fact, we conducted the experiments more than three times at a specific frequency. However, due to an oversight, we did not synchronize the experiments in the video with the corresponding pictures. Therefore, we have replaced the relevant movie.

Comment 23: The mucus clearing design looks very similar to the RoboCap design by Srinivasan, S. S. et al. RoboCap: Robotic mucus-clearing capsule for enhanced drug delivery in the gastrointestinal tract. *Sci. Robot.* 7, eabp9066 (2022) which you cited. What are the differences between the designs?

Response 23: Sorry for our unclear description. Yes, in this aspect of our work, we have referenced the function design of the RoboCap provided by Srinivasan et al., but there are obvious differences between these two studies. This aspect has been analyzed in the revised manuscript.

On page 13, we made revisions to the text:

“Oral drug delivery of large molecules is constrained by the degradation environment and malabsorption in the GI tract, e.g., peptides such as insulin and vancomycin have an oral bioavailability of less than 1%⁶⁰. To solve this problem, Traverso et al.²⁰ incorporated threaded features onto the outer surface of the capsule and introduced a motor-drive rotation to locally clear the mucus layer. This design innovation offers an effective mechanism for enhancement of drug absorption. Inspired by this, we implement similar functionality by developing a mucus-clearing MagCap with a threaded shell. It is noted that, the rotation of the capsule reported by Traverso et al.²⁰ is triggered in response to the pH change, necessitating no additional manipulation, thus constituting its most notable advantage, which shows great convenience and implementation in future applications. However, subject to its trigger mode, it is difficult to achieve the action at a specific site or a specific time. In contrast, magnetically actuated method needs to rely on an external magnetic field system, but can bring about a substantial enhancement of controllability.”

Comment 24: Line 342 – 343 – “...the absorbance in solution increases by 239.3% (absorption peak at 290 nm) and 318.5% (absorption peak at 330 nm)...” Why did you choose to compare absorbance at 290 nm and 330 nm?

Response 24: In Giovanni’s study²⁰, it is mentioned that “the collected sample was then assessed with absorbance spectroscopy at 330 nm, where higher absorbance indicated a greater concentration of mucus displaced from the SI lining.” However, during our testing process, we found that the absorbance at about 290 nm exhibited peak values. Therefore, we simultaneously retained both sets of data as support for the conclusion in the previous manuscript. With your reminder, we believe that this approach might lead to misunderstandings. Therefore, in the revised manuscript, we chose to compare absorbance at 290 nm and provided the absorbance curves in both cases (refer to revised Fig. 7d-ii).

Fig. 7d(ii) comparative absorbance curves of the two capsules after 10 minutes of rotation in the lining of the small intestine.

Comment 25: Capsule localization was very briefly discussed and perhaps you can cite some relevant papers.

Response 25: Thank you for your helpful suggestion. In the revised manuscript, **we have cited some relevant papers when discussing the capsule localization.** The main modifications are as follows.

On Page 17, we made revisions to the text:

“Lastly, it is favorable to know the position and orientation of the capsule because it

can guide the application of magnetic field for the desired motion, a key factor for this was the introduction of real-time localization techniques⁶⁶. Considering the medical CT imaging methods usually exist radiation for patients and health-care workers, we can account for some assisted positioning technologies for the long-term dynamic positioning of MagCaps in the GI environment, such as ultrasound⁶⁷ and magnetic^{68, 69} assisted localization technology.”

Comment 26: How would you turn the capsule around the bend during locomotion? What fields in the xyz directions were applied?

Response 26: Sorry for our unclear description. This point you mentioned is crucial in understanding the dynamics of the capsule. With your prompting, to better illustrate how the capsule turns within a 3D magnetic field, **we have extensively analyzed the capsule's movement in the revised manuscript** (please see newly added Text S4, Fig. S14, and Fig. S15).

Comment 27: Movie S9 – The capsule appeared to be stuck in mucus. Would higher frequency have helped move the capsule more effectively? What are your frequency limits before drug delivery is activated?

Response 27: Yes, appropriately increasing the frequency indeed helps improve the capsule's movement, as evident from Figure 3b. However, we've also observed that excessively high frequencies can lead to reduced movement efficiency. We believe that augmenting the magnetic field intensity might be feasible, although caution is necessary not to exceed the magnetic field required for the capsule valve to open.

Regarding the second question, our current experiments haven't specifically evaluated the capsule's movement efficiency under certain conditions. Nevertheless, we restrict the frequency based on the maximum rotation speed achievable by the robotic arm before drug delivery is initiated. We hope this explanation addresses your inquiries satisfactorily.

Comment 28: How well does the capsule perform under external forces from the body?

Response 28: Thank you for your valuable feedback. We posit that external forces exerted by the body primarily manifest as oscillations and pressure variances affecting

MagCaps. Considering both scenarios tend to favor capsule opening, we are particularly focused on the impact of these external forces on the capsule's sealing performance, as you've highlighted. Based on the experimental data in Figure 2, whether through rapid oscillation tests (Figure 2f-g) or the capsule pressure test (Figure 2e, a newly added test in the revised manuscript), it is evident that when the magnetic frame mass fraction exceeds 50%, the capsule can withstand interference from the external environment.

Comment 29: While multifunctionality is an interesting topic for wireless capsules, what are the author's thoughts of having one capsule with many functions versus many capsules with specialized functions? Would this depend on application?

Response 29: Thank you for your interest. Based on our understanding, wireless capsules, with the potential for multifunctionality, are typically capable of either housing multiple functions within a single capsule or being utilized as multiple specialized capsules for specific needs. In fact, in our work, we think that the highlighted multifunctionality in our paper encompasses these two aspects you mentioned. Throughout our manuscript, we delineate six categories of capsule structures where functionalities overlap to some extent while maintaining distinctiveness. **To reinforce this point, we have included a new table (Table S2) in the Supplementary Materials, presenting the distinctive functions of these six capsules.** This table can illustrate the concept of 'many capsules with specialized functions' as you mentioned. Simultaneously, within each category of capsules, there exists diversity compared to traditional capsules. This includes features such as active motion, integration of drug delivery and sampling functionalities, aligning with the concept of "having one capsule with many functions".

Based on this analysis, our intent is to showcase the potential versatility of the capsules we've designed, tailored to specific practical applications. The decision to adopt a certain approach relies heavily on real-world application scenarios, aiming to maximize the demonstrated capabilities of our capsule designs. **This point has been highlighted in the section of Discussion in the revised manuscript.**

REVIEWERS' COMMENTS

Reviewer #2 (Remarks to the Author):

Line 81 - "It's worth noting that, through innovations in capsule structure and 82 materials, some extremely small capsules or those driven by low magnetic fields have been developed⁴³⁻⁴⁵" – a major factor limiting the development of small capsules is not the actuation method but the reliance on batteries.

Line 97 – "The MaCap s" should be MagCaps – keep terminology consistent , see also Line 97, 99, 101 vs. Line 91

Line 117 should be checked for grammar

Line 287 – ex-vivo would be a better description of the pigs stomach than in-vitro.

Line 322 should be ""in-vivo validation", I still have doubts about the relevance of using a rabbit model since the references cited were for micromotors not magnetic capsules or other capsule endoscope like technology.

Reviewer #3 (Remarks to the Author):

The authors should review the manuscript and videos for errors in spelling, grammar, and punctuation (e.g. video S4, spelling of scale bar, etc.). Otherwise, the authors were able to provide additional evidence, details, and discussion to support their claims and results.

REVIEWER COMMENTS

Reviewer #2 (Remarks to the Author):

Line 81 - “It's worth noting that, through innovations in capsule structure and 82 materials, some extremely small capsules or those driven by low magnetic fields have been developed⁴³⁻⁴⁵” – a major factor limiting the development of small capsules is not the actuation method but the reliance on batteries.

Line 97 – “The MaCap s” should be MagCaps – keep terminology consistent , see also Line 97, 99, 101 vs. Line 91Line 117 should be checked for grammar

Line 287 – ex-vivo would be a better description of the pigs stomach than in-vitro.

Line 322 should be “in-vivo validation”, I still have doubts about the relevance of using a rabbit model since the references cited were for micromotors not magnetic capsules or other capsule endoscope like technology.

Reviewer #3 (Remarks to the Author):

The authors should review the manuscript and videos for errors in spelling, grammar, and punctuation (e.g. video S4, spelling of scale bar, etc.). Otherwise, the authors were able to provide additional evidence, details, and discussion to support their claims and results.

Responses to Comments on “NCOMMS-23-49636A”

Dear reviewers:

Thank you very much for your careful review of our manuscript. According to all comments, the manuscript has been revised carefully. We believe that the quality of our manuscript has been further improved.

The point-to-point responses to the comments are listed below.

Sincerely Yours,

Prof. Quanliang Cao & Prof. Liang Li

Wuhan National High Magnetic Field Center, Huazhong University of Science and
Technology

Response to the reviewer #2:

Comment 1: Line 81 - “It's worth noting that, through innovations in capsule structure and 82 materials, some extremely small capsules or those driven by low magnetic fields have been developed⁴³⁻⁴⁵” – a major factor limiting the development of small capsules is not the actuation method but the reliance on batteries.

Response 1: Thank you for your careful and professional comments. We agree with your viewpoint. The magnetic actuation method does not require a built-in battery (i.e. a built-in power circuit), which gives it an advantage over other methods in reducing capsule size. We have highlighted this information in the revision.

On Page 3, we made revisions to the text:

“Out of these methods, the magnetic actuation is more straightforward³⁶ and holds potential for size downscaling³⁷, as it eliminates the need for on-board control and power circuits.”

Comment 2: Line 97 – “The MaCap s” should be MagCaps – keep terminology consistent , see also Line 97, 99, 101 vs. Line 91

Response 2: Thank you for your reminder. We have carefully checked possible inconsistencies throughout the entire manuscript.

Comment 3: Line 117 should be checked for grammar

Response 3: We have checked and modified it.

Comment 4: Line 287 – ex-vivo would be a better description of the pigs stomach than in-vitro.

Response 4: Thank you for your suggestion. We have substituted all instances of "in-vitro" throughout the manuscript.

Comment 5: Line 322 should be “in-vivo validation”, I still have doubts about the relevance of using a rabbit model since the references cited were for micromotors not magnetic capsules or other capsule endoscope like technology.

Response 5: Thank you for your reminder. It should be “in-vivo validation”. The used

references are mainly used to show that similar drug delivery and release functions are provided in rabbit models. It is worth mentioning that Reference 51 is for magnetic capsules actually. It is reported that capsules of similar dimensions to those in our study have successfully demonstrated targeted transport and drug release in rabbit stomachs. Of course, it should be acknowledged that, as you said, it is more convincing to use larger animal models like pigs in the experiments, which closely resemble the human gastrointestinal environment. Therefore, we illustrate this point in the revision.

In the Conclusion section, we add:

“Simultaneously, it is worth noting that large animal models like pigs, with GI environments very similar to humans, should be prioritized in future studies.”

Response to the reviewer #3:

Comment 1: The authors should review the manuscript and videos for errors in spelling, grammar, and punctuation (e.g. video S4, spelling of scale bar, etc.). Otherwise, the authors were able to provide additional evidence, details, and discussion to support their claims and results.

Response 1: Gratitude for your meticulous peer review of our manuscript. A thorough examination of errors in the manuscript has been conducted.